# Failure-Driven Workflow Refinement

**Jusheng Zhang**[* 1 2] **Kaitong Cai**[* 1] **Jing Yang**[1] **Ziliang Chen**[1]
**Yongsen Zheng**[2] **Kwok-Yan Lam**[2] **Liang Lin**[1] **Keze Wang**[† 1]

## Abstract

Workflow optimization for tool-using LLM agents is often cast as global search over candidate graphs, scored by a scalar metric. This collapses rich, multi-step failure traces into binary outcomes, obscuring recurring failure structure and making refinement inefficient. We reframe optimization as *distributional refinement*: each workflow induces a density over a **Failure Signature Space** $\mathcal{F}$, and the goal is to minimize its **Expected Failure Mass**. We propose **CE-Graph**, which maintains a counterexample pool, estimates dense failure modes, and applies operator-constrained graph edits via a **Propose-and-Verify** loop with a convergence-aware stopping rule. Across math, code, and QA benchmarks, CE-Graph improves robustness while reducing optimization cost compared to strong workflow-search baselines, suggesting reliability emerges from learning and reshaping failure landscapes rather than merely maximizing aggregate success rates.

## 1. Introduction

Large Language Models (LLMs) (OpenAI et al., 2024; Vaswani et al., 2023; Zhao et al., 2025; Minaee et al., 2025) are increasingly used to power agentic workflows, where tasks are decomposed into multiple steps involving tool calls, logical control flows, and verification routines (Zhang et al., 2025e;a;f; Jeyakumar et al., 2024; Zhang et al., 2025c). Such workflows enable LLMs to handle long-horizon reasoning and complex problem-solving. The central challenge is how to optimize these workflows: given a dataset $D$, how can we construct a workflow $W$ that achieves the highest reliability?

This challenge is fundamentally distinct from traditional

program synthesis (Yu et al., 2025b; Sapkota et al., 2026). Unlike programs in a formal language, LLM-based workflows operate in a vast, unstructured space defined by natural language prompts and arbitrary tool interactions. Furthermore, their failures are not deterministic bugs but are often **stochastic and semantic** in nature, stemming from subtle flaws in the model's reasoning process (Steenhoek et al., 2025; Song et al., 2025; Huang et al., 2025; Zhang et al., 2025d;b). Consequently, established methods designed for formal program spaces are often ineffective, necessitating a new optimization paradigm.

A common formulation treats workflow optimization as a global program search (Zhang et al., 2025f;a; Li et al., 2024; Yu et al., 2025a). The objective is to find a configuration $W^*$ that maximizes a global, sample-based evaluation metric $G(W, D)$: $W^* = \arg\max_{W \in S} G(W, D)$. Recent systems exemplify this view by employing advanced search strategies such as Monte Carlo Tree Search (MCTS) (Coulom, 2006; Browne et al., 2012; undefinedwiechowski et al., 2022). While effective, this line of work inherits a core limitation rooted in its objective function, i.e., the reliance on **information collapse**. The rich, multi-step trace of each failure is compressed into a single binary signal, preventing the optimizer from modeling the **underlying failure distribution**, which is key to efficient refinement.

We argue that these failures are not random noise but are, in fact, samples from a workflow-specific failure distribution (Pan et al., 2025; Zhang et al., 2025g). This insight motivates a paradigm shift. We re-conceptualize workflow optimization not as maximizing a scalar score, but as a process of progressively reshaping this underlying failure distribution. We formalize this by introducing the **Failure Signature Space** $\mathcal{F}$ and casting the optimization goal as the minimization of the **Expected Failure Mass** $M(W)$: $W^* = \arg\min_{W \in S} M(W)$ where $M(W) = \int_{\mathcal{F}} p(\mathbf{s} \mid W) \, d\mathbf{s}$. This new formulation, which aims to minimize the integral of a failure density function, marks a fundamental departure from the conventional paradigm of maximizing a sample-based performance metric. Instead of treating the workflow as a black box, it provides a white-box and distributional lens. This allows us to move from inefficient, zero-order optimization to a principled, gradient-like descent directly

---

[1]Sun Yat-sen University, Guangzhou, China [2]Nanyang Technological University, Singapore. Correspondence to: Keze Wang <keze.wang@sysu.edu.cn>, Jusheng Zhang <jushengzhang1@gmail.com>.

*Proceedings of the 43$^{rd}$ International Conference on Machine Learning*, Seoul, South Korea. PMLR 306, 2026. Copyright 2026 by the author(s).

on the landscape of failure modes. To operationalize this principle, we propose CE-Graph (Counterexample-Guided Workflow Optimization), a framework that realizes this failure-mass-reduction strategy. It operates through three key stages. First, it maintains a counterexample pool and systematically diagnoses the densest regions of the failure distribution by clustering their **semantic signatures**, thereby approximating the "semantic gradient". Second, it replaces heuristic fixes with a principled **Propose-and-Verify** mechanism, which generates and empirically validates targeted graph edits designed to reduce the mass of these failure modes. Third, our CE-Graph incorporates a convergence-aware stopping rule to halt optimization once the failure distribution has stabilized.

We evaluate our CE-Graph on math, code, and QA benchmarks, showing that it achieves higher success rates at significantly lower optimization cost compared to strong workflow search baselines. Beyond these empirical gains, our work highlights a fundamental principle for building robust agentic systems: true reliability emerges not from attempting to avoid failures, but from systematically understanding and resolving their underlying distributional structure. Conceptually, CE-Graph establishes a **self-referential optimization paradigm**, employing language models to analyze and refine the failure distributions of the very LLM-based systems they compose.

## 2. A New Optimization Paradigm for LLM Workflows

We develop the theoretical foundation of CE-Graph by proposing a distributional paradigm for LLM workflow optimization. We introduce the **Failure Signature Space** and formulate optimization as minimizing the *expected failure mass* in this space (Ribeiro et al., 2020). Unlike scalar success metrics, this view preserves *where/how* failures occur, enabling mode-targeted refinement. We then formalize the dominant "global search" paradigm and show its intrinsic limitation: it observes only collapsed signals and is therefore blind to the geometry of failure distributions (Zhang et al., 2025f). This motivates our paradigm, **failure-driven refinement**, which iteratively reduces failure mass via mode-conditioned edits, setting the stage for the algorithm in Section 3.

### 2.1. Problem Formulation in Failure Signature Space

Execution traces of LLM workflows are high-dimensional and hard to compare directly. We map failures into a shared, structured space for systematic optimization. Let a workflow be $W \in S$, and let $\tau \sim \mathcal{D}_W$ denote an execution trace induced by running $W$. Define a failure indicator $Z(\tau) \in \{0, 1\}$ and a signature mapping $\phi\colon \mathbf{s} = \phi(\tau) \in \mathcal{F}$, where $\mathcal{F}$ is the **Failure Signature Space**.

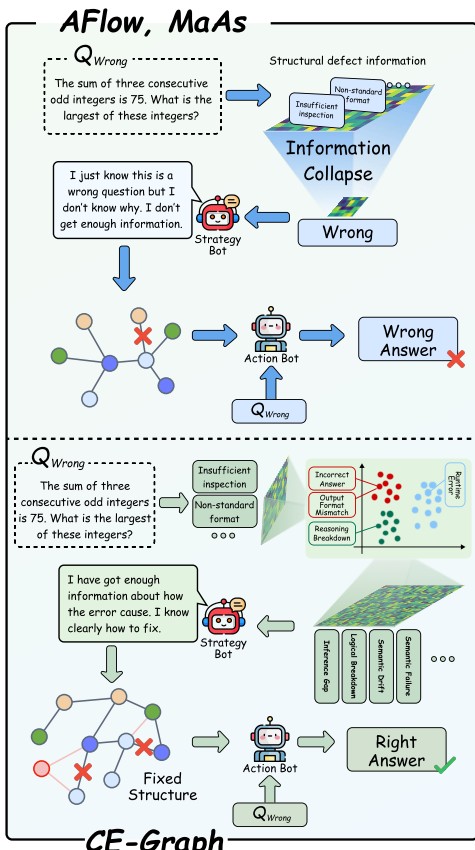

*Figure 1.* Upper: scalar-only optimization causes "information collapse." Lower: **CE-Graph** organizes failures in signature space and refines workflows by dominant modes.

**Unnormalized failure density.** A normalized conditional density over failures would integrate to 1 and cannot directly serve as an objective. We therefore define an *unnormalized* failure density:

$$\rho(\mathbf{s} \mid W) := \mathbb{P}(Z(\tau) = 1,\ \phi(\tau) \in d\mathbf{s} \mid W), \quad (1)$$

which represents the failure-probability mass assigned to regions of $\mathcal{F}$.

**Severity-weighted expected failure mass.** To account for heterogeneous failure severities, let $\omega(\tau) \geq 0$ be a severity weight (default $\omega \equiv 1$). We define:

$$W^* = \arg \min_{W \in S} M(W),$$
$$M(W) = \mathbb{E}_{\tau \sim \mathcal{D}_W}[Z(\tau) \cdot \omega(\tau)] = \int_{\mathcal{F}} \bar{\omega}(\mathbf{s})\, \rho(\mathbf{s} \mid W)\, d\mathbf{s} \quad (2)$$

where $\bar{\omega}(\mathbf{s}) := \mathbb{E}[\omega(\tau) \mid \phi(\tau) = \mathbf{s}, Z(\tau) = 1]$. When $\omega \equiv 1$, $M(W)$ reduces to failure probability, while $\rho(\mathbf{s} \mid W)$ still exposes failure *modes* in $\mathcal{F}$.

We assume (i) traces are i.i.d. samples from $\mathcal{D}_W$ for fixed $W$, (ii) $\phi$ captures relevant semantic/structural information,

and (iii) dominant modes of $\rho(\mathbf{s} \mid W)$ are estimable from counterexamples. Edits are discrete and the objective is non-convex; thus we use density-based signals to guide *mode-conditioned refinement* rather than claiming differentiable gradient descent.

## 2.2. The Pitfalls of Global Search

The dominant global search paradigm (e.g., MCTS in MaAS (Zhang et al., 2025a) / AFlow (Zhang et al., 2025f)) maximizes a black-box score $G(W, D)$ (approximately success rate on dataset $D$), i.e., $W^* = \arg\max_{W \in S} G(W, D)$. This suffers from **information collapse**: each trace is reduced to $\text{Observe}(\tau_d) \in \{0, 1\}$, discarding *where/why* it failed. In our formulation, the optimizer does not observe $\mathbf{s} = \phi(\tau_d)$ and thus cannot model the geometry of $\rho(\mathbf{s} \mid W)$ or isolate dominant failure modes, resulting in inefficient zero-order exploration (Snoek et al., 2012). Our paradigm instead explicitly discovers modes and targets them.

## 2.3. The Failure-Driven Refinement Paradigm

We reframe optimization as iterative refinement: $W_{t+1} = W_t \oplus \Delta_t$, inspired by counterexample-guided ideas but adapted to stochastic, semantic LLM failures (Debbi, 2018; Shorinwa et al., 2025). Since global minimization is intractable, we adopt a greedy objective:

$$
\begin{aligned}
\Delta_t &= \arg\max_{\Delta \in \mathcal{A}(W_t, \mathcal{O})} \left[ M(W_t) - M(W_t \oplus \Delta) \right] \\
&= \arg\max_{\Delta \in \mathcal{A}(W_t, \mathcal{O})} \left[ \int_{\mathcal{F}} \bar{\omega}(\mathbf{s}) \rho(\mathbf{s} \mid W_t) \, d\mathbf{s} - \int_{\mathcal{F}} \bar{\omega}(\mathbf{s}) \rho(\mathbf{s} \mid W_t \oplus \Delta) \, d\mathbf{s} \right]
\end{aligned}
\tag{3}
$$

This is a blueprint for approximation: we target dominant failure modes and apply operator-constrained discrete edits.

### 2.3.1. THEORETICAL PROPERTIES OF GREEDY MASS REDUCTION

Assume $\rho(\mathbf{s} \mid W)$ is Lipschitz with constant $L$ and edits have bounded impact $\|\rho(\mathbf{s} \mid W \oplus \Delta) - \rho(\mathbf{s} \mid W)\|_\infty \leq B$. Let $b_t^* \subseteq \mathcal{F}$ be the targeted mode.

**Theorem 2.1** (Greedy Reduction Bound). *If the edit reduces the (severity-weighted) mass inside $b_t^*$ by at least $\delta > 0$, then $M(W_{t+1}) \leq M(W_t) - \delta + \epsilon$, where $\epsilon = O(L \cdot B \cdot \mu(\mathcal{F} \backslash b_t^*))$ bounds spillover to non-target regions.*

*Proof Sketch.* Split the integral in Eq. 3 into $b_t^*$ and its complement. The targeted reduction gives $-\delta$ on $b_t^*$, and Lipschitz plus bounded edit impact yields the spillover term $\epsilon$. Full details are in Appendix G. □

---

**Algorithm 1** The CE-Graph Algorithm

1: **Input:** Initial workflow $W_0$, Dataset $D$, Hyperparameters $N, K, k, \epsilon, T_{\max}$
2: **Output:** Optimized workflow $W_T$
3: Initialize $t \leftarrow 0$, validation scores $\mathcal{G} \leftarrow \emptyset$
4: **while** not converged **and** $t < T_{\max}$ **do**
5:   *// Sample failures induced by $W_t$ and build an empirical estimate of $\rho(\mathbf{s} \mid W_t)$*
6:   Populate Counterexample Pool $C_t$ by executing $W_t$ on $D$.
7:   **if** $C_t$ is empty **then**
8:     **break** *// No failures; optimization complete*
9:   **end if**
10:   Construct embeddings $S_t = \{\phi(\tau_d) \mid (d, \tau_d) \in C_t\}$.
11:   Optionally compute severity weights $\Omega_t = \{\omega(\tau_d) \mid (d, \tau_d) \in C_t\}$ (default: $\omega \equiv 1$).
    *// Approximate greedy mass reduction (Eq. 3)*
12:   **// Step 1: Target a dominant failure mode**
13:   Fit GMM to $S_t$ to obtain modes $B_t$ and parameters $\theta$.
14:   Identify the most consequential mode $b_t^*$ using Eq. 5.
15:   **// Step 2: Find an effective operator-constrained edit**
16:   Propose candidates $\{\Delta_1, \ldots, \Delta_N\} \subset \mathcal{A}(W_t, \mathcal{O})$ conditioned on $b_t^*$.
17:     Select $\Delta_t \leftarrow \arg\max_{\Delta_i} V(\Delta_i)$ via Monte Carlo verification.
18:     Apply: $W_{t+1} \leftarrow W_t \oplus \Delta_t$.
    *// Evaluation and check*
19:     Evaluate $g_t$ on validation set; update $\mathcal{G}$.
20:   $t \leftarrow t + 1$
21: **end while**
22: **return** $W_t$

---

## 3. CE-Graph: A Practical Algorithm for Failure Mass Reduction

We detail our CE-Graph as a practical approximation to the intractable greedy failure mass reduction objective (Eq. 3) (Clarke et al., 2000), transforming abstract concepts into actionable steps. By deconstructing the challenge into three principled stages—**(i) empirical failure modeling in signature space**, **(ii) dominant-mode targeting**, and **(iii) propose-and-verify operator-constrained edits**—we operationalize a *mode-conditioned discrete refinement* on the failure landscape (Bishop, 2006). This not only approximates the theoretical ideal but also addresses real-world challenges such as stochastic failures and large edit spaces (Zhang et al., 2025f). The following subsections elaborate on each stage's role, including a detailed analysis of their approximations and alignments with the theory.

### 3.1. Constructing the Failure Signature Space $\mathcal{F}$

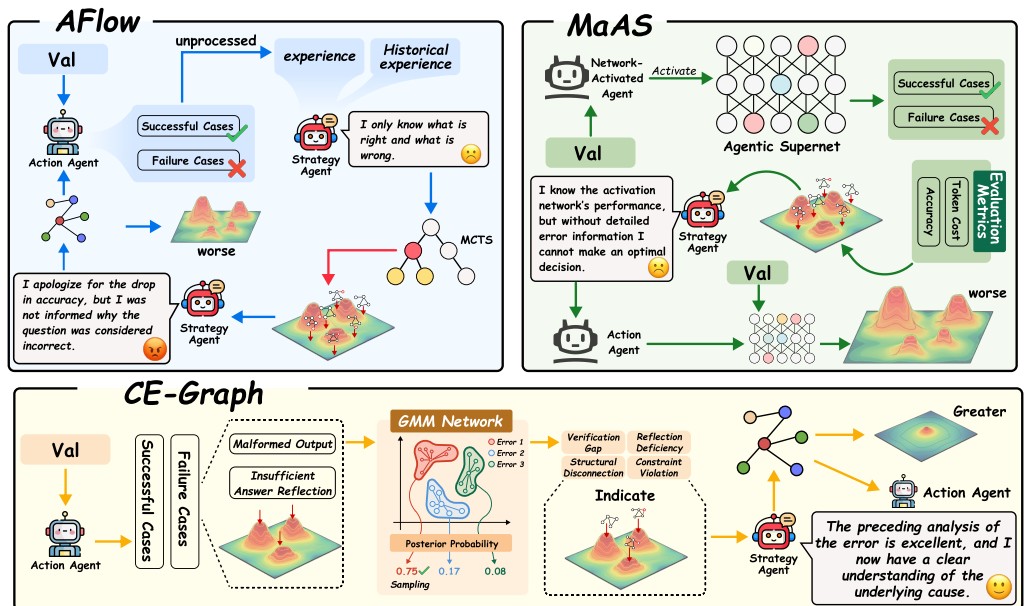

*Figure 2.* **Overview of CE-Graph.** (i) distill failure traces into signatures; (ii) identify dominant failure modes; (iii) propose-and-verify graph edits to reduce failure mass.

The first step transforms unstructured data in the Counterexample Pool $C_t$ into a tractable format, enabling empirical modeling of $\rho(\mathbf{s} \mid W_t)$. Without this, failures remain opaque, as in global search methods. We define a mapping $\phi$ that projects a raw trace $\tau_d$ into the structured **Failure Signature Space** $\mathcal{F}$ through two stages, ensuring both structural and semantic fidelity.

First, **Failure Distillation** uses a utility LLM to distill $\tau_d$ into a concise tuple $(v_{\mathrm{err}}, z_{\mathrm{err}})$, capturing the error-producing node and message. This step compresses verbose traces (e.g., a multi-step math failure involving incorrect LCM calculation) into analyzable components. Second, **Semantic-Structural Vectorization** maps this tuple to the signature vector $\mathbf{s}$, preserving orthogonality and similarity.

Formally, let $V$ be the set of workflow graph nodes and $\mathcal{Z}$ the space of error messages. We define: - A **structural mapping** $\psi_{\mathrm{struct}} : V \to \mathbb{R}^{|V|}$, yielding a one-hot vector for a node identifier. - A **semantic mapping** $\psi_{\mathrm{sem}} : \mathcal{Z} \to \mathbb{R}^d$, embedding an error message into $d$ dimensions (e.g., using BERT-like models for semantic clustering).

The final failure signature is:

**Definition 3.1** (Failure Signature Vector)**.** A failure signature $\mathbf{s} \in \mathcal{F}$ for a distilled trace $(v_{\mathrm{err}}, z_{\mathrm{err}})$ is:

$$\mathbf{s} = \phi(\tau_d) = \psi_{\mathrm{struct}}(v_{\mathrm{err}}) \oplus \psi_{\mathrm{sem}}(z_{\mathrm{err}}), \qquad (4)$$

where $\oplus$ denotes vector concatenation.

The structural component $\psi_{\mathrm{struct}}(v_{\mathrm{err}})$ identifies *where* the failure occurred, mapping to orthogonal subspaces in $\mathcal{F}$ for node-specific patterns, crucial for attributing blame in

multi-agent workflows. The semantic component $\psi_{\mathrm{sem}}(z_{\mathrm{err}})$ captures *what* error occurred, grouping semantically similar messages (e.g., "calculation error" vs. "the sum was incorrect"). This hybrid embedding endows the point cloud $S_t = \{\phi(\tau_d) \mid \tau_d \in C_t\}$ with an informative geometry for approximating dominant modes of $\rho(\mathbf{s} \mid W_t)$, reducing information loss and exposing failure clusters that global methods miss.

## 3.2. Solving the Refinement Objective

CE-Graph approximates the greedy mass reduction by two steps: (i) targeting a dominant failure mode, then (ii) finding an effective edit. This decomposition balances computational tractability with theoretical fidelity, ensuring each iteration aims to reduce $M(W_t)$ as per Theorem 2.1.

### 3.2.1. STEP 1: TARGETING A DOMINANT FAILURE MODE

We approximate the greedy objective by identifying a dominant region of the empirical failure landscape—a mode that contributes the most to the current failure mass. Concretely, we fit a Gaussian Mixture Model (GMM) with parameters $\theta$ to the signature set $S_t$. Each component $b_k$ represents a distinct failure mode (e.g., a cluster of division-by-zero errors in code tasks), with prevalence $\pi_k$. To align with the (possibly severity-weighted) objective in Eq. 2, we select the mode with the largest *estimated mass contribution*:

$$b_t^* = \underset{b_k \in B_t}{\arg\max} \, \widehat{m}_k, \qquad \widehat{m}_k := \sum_{\mathbf{s}_i \in S_t} p(b_k \mid \mathbf{s}_i, \theta) \cdot \omega(\tau_i),$$

$$(5)$$

where $p(b_k \mid \mathbf{s}_i, \theta)$ is the posterior responsibility and $\omega(\tau_i)$ is the (optional) severity weight of the corresponding counterexample trace. When $\omega \equiv 1$, this reduces to selecting the most prevalent mode (largest effective $\pi_k$), recovering the standard "densest cluster" criterion.

Intuitively, $b_t^*$ serves as a **mode-conditioned direction**: it identifies a systematic weakness to prioritize, so that subsequent edits address concentrated failure mass rather than isolated outliers.

### 3.2.2. ANALYSIS OF MODE TARGETING APPROXIMATION

The GMM provides a consistent estimator of the dominant structure in the empirical signature distribution under standard conditions, making it a practical proxy for the true (unnormalized) failure density $\rho(\mathbf{s} \mid W_t)$. As $|S_t|$ grows, the estimated responsibilities concentrate around coherent modes, and the selected $b_t^*$ increasingly reflects the region with the largest contribution to $M(W_t)$. Importantly, this step is *not* a differentiable gradient computation: workflow edits are discrete, and our goal is to obtain a high-impact target region for refinement. In benchmarks like MATH (Hendrycks et al., 2021), where failures cluster around recurring algebraic misconceptions, mode targeting effectively identifies "mountains" whose reduction yields outsized improvements, compared to uniform sampling that spreads budget across low-impact regions.

### 3.2.3. STEP 2: FINDING THE OPTIMAL EDIT ($\Delta_t$) VIA PROPOSE-AND-VERIFY

This step addresses two challenges: the vast edit space $\mathcal{A}(W_t, \mathcal{O})$ (potentially exponential in workflow size) and the unknown utility distribution over instances. Our **Propose-and-Verify** mechanism handles them in two stages, combining LLM creativity with empirical validation to approximate the argmax in Eq. 3.

**Constrained Proposal:** A "Proposer" LLM, conditioned on $b_t^*$'s summary (e.g., "recurring off-by-one errors in loops"), generates $N$ diverse high-potential edits $\{\Delta_1, \ldots, \Delta_N\}$, reducing the search to a promising subset. This heuristic leverages LLM's generative power while constraining outputs to the operator library $\mathcal{O}$, ensuring feasibility and keeping the refinement step local.

**Verification:** To align with mass reduction, we evaluate each candidate on $K$ sampled instances associated with the target mode $b_t^*$ and estimate its post-edit failure mass on that mode. Let $\tau_k^{(i)}$ be the trace obtained by executing $W_t \oplus \Delta_i$ on sampled instance $x_k$ (from the counterexamples assigned to $b_t^*$), and let $Z(\tau_k^{(i)})$ indicate failure under the verifier. We

define:

$$V(\Delta_i) \approx -\frac{1}{K} \sum_{k=1}^{K} Z\left(\tau_k^{(i)}\right) \cdot \omega\left(\tau_k^{(i)}\right), \qquad (6)$$

so that larger $V(\Delta_i)$ corresponds to smaller estimated failure mass on the target mode. The highest $V(\Delta_i)$ yields $\Delta_t$, enabling efficient, evidence-based refinement. This Monte Carlo estimator directly tracks the quantity our theory aims to reduce (Eq. 2), with variance decreasing as $O(1/K)$ under bounded $\omega$, aligning with the spirit of Theorem 2.1. Further details are provided in Appendix H.

## 4. Experiments

We benchmark our *failure-driven refinement* paradigm against a comprehensive suite of baselines across mathematical reasoning, code generation, and complex tool-use tasks, to evaluate both (i) final-task performance and (ii) optimization efficiency.

### 4.1. Experimental Setup

**Tasks and Benchmarks.** We evaluate CE-Graph on public benchmarks spanning three domains: **(1) Math reasoning:** GSM8K (Cobbe et al., 2021), MATH (Hendrycks et al., 2021), and MultiArith (Roy & Roth, 2015); **(2) Code generation:** HumanEval and MBPP (Chen et al., 2021; Austin et al., 2021); **(3) Tool use:** GAIA (Mialon et al., 2023). For GSM8K/MATH/MultiArith, we report exact-match accuracy; for HumanEval/MBPP, we report `pass@1` under the standard evaluation harness. For GAIA, we follow the official Level 1–3 protocol and report accuracy per level and the overall average. **Baselines.** We compare against three categories: **Single-agent execution** (Vanilla, CoT (Wei et al., 2022), ComplexCoT (Kojima et al., 2022), and Self-Consistency (Wang et al., 2023)); **Hand-crafted multi-agent systems** (MultiPersona (Wang et al., 2024), LLM-Debate (Du et al., 2024), LLM-Blender (Jiang et al., 2023), DyLAN (Liu et al., 2024b), AgentVerse (Chen et al., 2024b), MacNet (Qian et al., 2025)); **Automated agentic systems** (AutoAgents (Chen et al., 2024a), GPTSwarm (Zhuge et al., 2024), ADAS (Hu et al., 2025), AgentSquare (Shang et al., 2025), AFlow (Zhang et al., 2025f), and MaAS (Zhang et al., 2025a)). Unless a baseline defines a mandatory controller, we standardize the execution environment (same tool access when applicable) to ensure comparability.

**Base LLM and Decoding.** All methods use the same base model, `gpt-4o-mini` (OpenAI et al., 2024), to avoid conflating algorithmic gains with model capacity. We use deterministic decoding (temperature $= 0$) for single-run methods. For Self-Consistency (CoT×5), we follow the standard setting with 5 independent samples and majority vote. For methods that require multiple agents, each agent is instanti-

*Table 1.* A complete performance comparison on math and code benchmarks. Results are highlighted using shades of gray for the header and top performers, bold for the best, and underlining for the runner-up.

| Method | GSM8K | MATH | MultiArith | HumanEval | MBPP | Avg. |
|---|---|---|---|---|---|---|
| *Single Agent Execution* | | | | | | |
| Vanilla (Baseline) | 87.45 | 46.29 | 96.85 | 87.08 | 71.83 | 77.50 |
| CoT (Wei et al., 2022) | 87.10 | 46.40 | 96.31 | 88.13 | 71.83 | 77.95 |
| ComplexCoT (Kojima et al., 2022) | 86.89 | 46.53 | 96.70 | 87.49 | 72.36 | 78.00 |
| SC (Wang et al., 2023) (CoT×5) | 87.57 | 47.91 | 96.58 | 88.60 | 73.60 | 78.85 |
| *Hand-craft Multi-agent Systems* | | | | | | |
| MultiPersona (Wang et al., 2024) | 87.50 | 45.43 | 97.49 | 88.32 | 73.19 | 78.39 |
| LLM-Debate (Du et al., 2024) | 89.47 | 48.54 | 97.33 | 88.68 | 70.29 | 78.86 |
| LLM-Blender (Jiang et al., 2023) | 88.35 | 46.92 | 97.29 | 88.80 | 77.05 | 79.68 |
| DyLAN (Liu et al., 2024b) | 89.98 | 48.63 | 97.12 | 90.42 | 77.30 | 80.69 |
| AgentVerse (Chen et al., 2024b) | 89.91 | 47.35 | 97.50 | 89.29 | 74.28 | 79.67 |
| MacNet (Qian et al., 2025) | 87.95 | 45.18 | 96.03 | 84.57 | 65.28 | 75.00 |
| *Automated Agentic Systems* | | | | | | |
| AutoAgents (Chen et al., 2024a) | 87.69 | 45.32 | 96.42 | 87.64 | 71.95 | 77.80 |
| GPTSwarm (Zhuge et al., 2024) | 89.14 | 47.88 | 96.79 | 89.32 | 77.43 | 80.11 |
| ADAS(Hu et al., 2025) | 86.12 | 43.18 | 96.02 | 84.19 | 68.13 | 75.13 |
| AgentSquare(Shang et al., 2025) | 87.62 | 48.51 | 97.77 | 89.08 | 78.46 | 80.29 |
| AFlow(Zhang et al., 2025f) | 91.16 | 51.28 | 96.22 | 90.93 | 81.67 | 82.25 |
| MaAS(Zhang et al., 2025a) | 92.30 | 51.82 | 98.80 | 92.85 | 82.17 | 83.59 |
| **CE-Graph (Ours)** | **93.70** | **55.91** | **99.20** | **94.26** | **88.10** | **86.23** |

ated as an independent call to the same base LLM, with role prompts specified by the method. A key concern in agentic optimization is whether gains come from *larger budgets* rather than better algorithms. To ensure strict comparability, we enforce **budget parity** at two levels:

**(i) Deployment-time inference budget (per test instance).** Across all methods, we impose the same hard caps on resource usage: a maximum number of LLM calls, a maximum total token budget (prompt+completion), and (when applicable) a maximum number of tool/verifier calls. Multisample or multi-agent baselines (e.g., SC, Debate, DyLAN) must allocate within the same global caps, rather than exceeding them via extra agents/samples. We report the *realized* inference-time tokens and costs in our cost analysis (Fig. 3, right), so improvements cannot be attributed to larger per-instance budgets. **(ii) Offline optimization budget (for automated workflow search/refinement).** For automated agentic systems (AutoAgents/GPTSwarm/ADAS/AgentSquare/AFlow/MaAS) and CE-Graph, we additionally control the total optimization budget: the total number of candidate workflow evaluations and the total optimization tokens consumed during search/refinement are fixed to the same cap. Concretely, each method is run under the same maximum number of refinement/search iterations and the same validation protocol, and is stopped early only if it meets its own convergence criterion within the shared budget. Thus, the final workflow used in Table 1 is the best one

found under an equal optimization budget, not an unequal search cost.

**CE-Graph Initialization, Optimization Budget, and Operators.** CE-Graph starts from a minimal initial workflow $W_0$ (a single solve node plus a verifier) and iteratively refines it by counterexample-guided local graph edits. We run optimization for up to 20 iterations and stop early if validation improvement saturates under our convergence-aware criterion (Section 3). Our operator library $\mathcal{O}$ contains three executable edits: `RevisePrompt` (prompt repair for an existing node), `InsertNode` (add a new sub-step with explicit I/O), and `DeleteNode` (remove redundant/degenerate steps). Each proposed edit is validated via a propose-and-verify stage; only edits that improve validation metrics are committed to update $W_{t+1}$.

### 4.2. Performance Analyses

As shown in Table 1, CE-Graph consistently outperforms all baselines across every task domain and establishes a new state-of-the-art average score of **86.23%**, widening the gap over the previous best of 83.59% achieved by MaAS. The gains are substantial across categories: on the **MATH** benchmark, CE-Graph improves accuracy by +4.1%; on the **MBPP** code synthesis task, it surpasses MaAS by +5.9%; and on **HumanEval**, it delivers an additional +1.4% improvement. Together, CE-Graph demonstrates balanced

improvements across mathematical reasoning, program synthesis, and tool-use domains. Figure 3 highlights the advantages of our failure-driven paradigm in complex, multi-domain environments like GAIA. Unlike monolithic systems (e.g., AFlow, ADAS) that struggle to cover heterogeneous task requirements with a static workflow, CE-Graph iteratively repairs prevalent failure modes, evolving into a robust workflow capable of cross-domain adaptation. This drives substantial gains across all levels—especially Level 3—underscoring its general-purpose capability. Furthermore, CE-Graph exhibits remarkable consistency across math, code, and tool-use. Crucially, its advantage widens with difficulty: while baselines plateau on long-horizon problems, CE-Graph maintains steady gains by systematically resolving structural weaknesses in the failure landscape.

### 4.3. Cost Analyses

Beyond raw performance, we analyze both **deployment-time** and **optimization-time** costs. For deployment-time efficiency, Fig. 3 (right) reports the realized token usage and API cost under the same inference budget caps for all methods, highlighting CE-Graph's favorable accuracy–efficiency trade-off. For optimization-time efficiency, CE-Graph reduces redundant trial-and-error by targeting dominant failure modes and validating only a small set of operator-constrained edits, leading to lower optimization token consumption under an equal search/refinement budget. Moreover, our convergence-aware stopping criterion prevents unnecessary iterations once validation improvements saturate, yielding substantial savings compared to fixed-iteration optimization. Ablations in Section 4.5 further confirm that clustering, verification, and structured operators are all critical for achieving strong performance *without* increasing budget.

### 4.4. Clustering-Driven Correction Stability

To precisely evaluate the stability and sustained refinement capability of our CE-Graph, we design a longitudinal assessment aimed at verifying its long-term effectiveness in locating and resolving fundamental failures. We begin by executing an unoptimized baseline workflow ($W_0$) on three mathematical reasoning benchmarks, i.e., GSM8K, MATH, and MultiArith, and collect all resulting failure cases to construct a fixed initial failure set ($E_0$). During the subsequent 20 rounds of optimization, each updated workflow ($W_t$) is re-evaluated on this fixed set ($E_0$). We continuously track a single core metric: *Accuracy on $E_0$*, defined as the proportion of initial failure examples successfully repaired by $W_t$. This fixed-set evaluation design effectively isolates variables, thereby providing a clear lens through which to assess the refinement mechanism's inherent capacity for stable and accumulative improvements over time. The lon-

*Table 2.* Ablation of CE-Graph on MATH and HumanEval. We introduce *w/ Generic Operators* to verify automation capabilities without domain-specific engineering. Metrics: Acc./pass@1 (%); Cost: optimization tokens ($\times 10^3$).

| Method | MATH | | HumanEval | |
|---|---|---|---|---|
| | Acc. | Cost | pass@1 | Cost |
| **CE-Graph (Full)** | **55.91** | **1210** | **94.26** | **955** |
| w/o Clustering | 51.25 | 1190 | 91.50 | 940 |
| w/o Verification | 49.10 | 1250 | 89.75 | 980 |
| w/ Generic Operators | 53.40 | 1290 | 92.10 | 1020 |
| w/o Structured Operators | 47.35 | 1310 | 87.20 | 1050 |
| w/o Conv.-aware Stopping | 53.50 | 1850 | 93.80 | 1520 |

gitudinal evaluation conducted on fixed failure sets ($E_0$) across the three mathematical reasoning benchmarks clearly demonstrates the superior stability and sustained improvement capability of our CE-Graph. As shown in Figure 4, CE-Graph achieves a smooth, monotonic trajectory, avoiding the *policy oscillation* and *information collapse* typical of baselines like AFlow. Through failure clustering and principled verification, it transforms discrete signals into consistent, net-positive iterations.

### 4.5. Ablation Studies

To validate CE-Graph's components and automation capability, we evaluate five variants: (1) **w/o Clustering**: excludes failure pattern identification; (2) **w/o Verification**: omits the validation step; (3) **w/ Generic Operators**: restricts actions to fundamental edits (e.g., *Add*, *Modify*) without domain priors to test efficacy under minimal human intervention; (4) **w/o Structured Operators**: replaces the library with free-form editing; and (5) **w/o Conv.-aware Stopping**: removes adaptive stopping. We analyze performance (Acc., pass@1, cost) and optimization dynamics on MATH and HumanEval to isolate each module's impact.

Table 2 validates CE-Graph's component synergy. **w/ Generic Operators** maintains competitive performance (MATH: 53.40%, HumanEval: 92.10%), proving robust automation even without domain priors. In contrast, the sharp drop in **w/o Structured Operators** (MATH: 47.35%) confirms the necessity of discrete action spaces for executable repairs. **w/o Verification** and **w/o Clustering** highlight the critical roles of hallucination filtering and mode-targeting, respectively. Finally, **w/o Convergence-aware Stopping** spikes costs by $> 50\%$, demonstrating its essential role in efficiency.

## 5. Related Works

**Automated Optimization of Agentic Workflows** Recent systems (e.g., MaAS (Zhang et al., 2025a), AFlow (Zhang et al., 2025f), AgentSquare (Shang et al., 2025), Adapt-Flow (Zhu et al., 2025)) optimize agentic workflows via global search (MCTS/evolution) to maximize success rates.

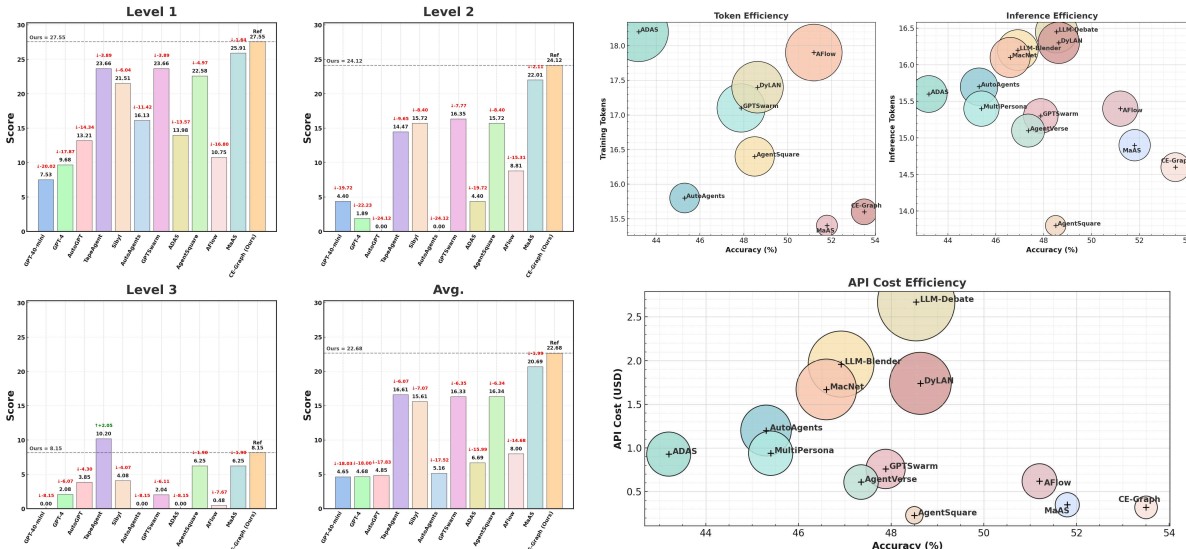

Figure 3. **CE-Graph** outperforms all baselines on GAIA (Levels 1–3, Avg.) and achieves the best accuracy–efficiency trade-off (tokens, API cost) in the ideal lower-right region.

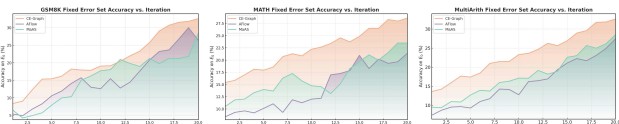

Figure 4. Refinement accuracy on a fixed failure set ($E_0$) over 20 rounds on GSM8K, MATH, and MultiArith. **CE-Graph** improves monotonically; baselines (e.g., **AFlow**) fluctuate.

Prompt optimizers such as DSPy (Khattab et al., 2024) also rely on scalar rewards, but this can induce "information collapse" by compressing rich failure traces into coarse signals, leading to expensive zero-order exploration. CE-Graph instead preserves semantic and structural failure information in a Failure Signature Space and optimizes via Expected Failure Mass minimization, enabling targeted, cost-effective repairs and improved robustness.

**Counterexample-Guided Refinement in Formal Methods** Counterexample-guided refinement, central to program synthesis and verification (e.g., CEGAR (Hidvégi et al., 2024), CigaR (Renze & Guven, 2024)), iteratively fixes models using violating examples. However, classical methods are designed for symbolic/deterministic settings and do not directly handle the stochastic, semantic failures of LLM workflows (Chang et al., 2024). CE-Graph adapts the principle to open-ended LLM domains by replacing formal checking with empirical failure-mode clustering and constrained graph edits, improving reliability beyond what symbolic checkers can capture.

**Instance-Level Self-Correction and Reflection** Instance-level self-correction methods (Self-Consistency (Wang et al., 2023), Reflexion (Shinn et al., 2023), Self-Refine (Madaan et al., 2023)) improve single outputs via voting or reflection, and are studied in agent benchmarks such as Agent-Bench (Liu et al., 2024a). While effective on individual cases, they often miss systemic failure patterns and can be unstable across iterations. CE-Graph takes a distributional view: it clusters counterexamples into recurring modes and applies structured repairs to fix root causes, yielding more stable and scalable refinement across domains.

## 6. Conclusion

We select CE-Graph as the optimization framework for agentic workflows because it directly addresses a core limitation of prior approaches: the collapse of rich failure traces into scalar rewards. By modeling failures in a structured signature space and explicitly identifying recurring failure modes, CE-Graph enables targeted, mode-level refinement rather than heuristic or search-heavy adjustments. The Propose-and-Verify mechanism further ensures that each edit is both constrained and empirically validated, yielding stable improvements without incurring the high cost and instability of global search.

## Impact Statement

Our work advances the field of automated agent design by introducing a resource-efficient optimization framework, CE-Graph. By focusing on distributional failure analysis rather than brute-force search, we drastically lower the computa-

tional cost required to build robust agentic systems. While this improves the accessibility and reliability of autonomous agents, it also underscores the need for responsible deployment. As agents become more capable of self-correction and complex reasoning, ensuring these capabilities are aligned with human values and safety standards remains a critical priority for the community.

## Acknowledgments

This work was supported in part by the National Natural Science Foundation of China (NSFC) under Grant 62276283; in part by the China Meteorological Administration's Science and Technology Project under Grant CMAJBGS202517; in part by the Guangdong-Hong Kong-Macao Greater Bay Area Meteorological Technology Collaborative Research Project under Grant GHMA2024Z04; in part by the Fundamental Research Funds for the Central Universities, Sun Yat-sen University, under Grants 23hytd006 and 23hytd006-2; in part by the Guangdong Provincial High-Level Young Talent Program under Grant RL2024-151-2-11; and in part by the National Research Foundation, Singapore, and Infocomm Media Development Authority under its Trust Tech Funding Initiative. Any opinions, findings, conclusions, or recommendations expressed in this material are those of the author(s) and do not reflect the views of the National Research Foundation, Singapore, or Infocomm Media Development Authority.

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

# A. Implementation Details for Failure Diagnosis

The efficacy of the CE-GRAPH framework is predicated on the fidelity of its initial diagnosis. Consequently, the reliability of the **Failure Distillation** step (Section 3.1) is paramount. We engineer architectural resilience through two primary mechanisms that exploit the discriminatory strengths of LLMs while mitigating their generative stochasticity.

**1. Constrained Task Formulation.** We deliberately eschew open-ended generation for diagnosis, as high-entropy sampling is inherently susceptible to hallucination. Instead, we reformulate failure distillation as a **low-entropy, discriminative extraction task**. The utility LLM is prompted to operate strictly as a parser, localizing the fault to a specific node within the known workflow topology and extracting a concise error signature from the trace. By constraining the output space to structured classifications rather than free-form reasoning, we leverage the robust instruction-following capabilities of modern models to ensure high-fidelity diagnostics.

**2. Statistical Fault Tolerance.** Our framework adopts a probabilistic view of diagnosis, adhering to a design philosophy of statistical robustness rather than requiring zero-shot perfection. Instances where the utility LLM yields low-confidence or unparseable outputs are flagged as "undiagnosable" and temporarily masked from the clustering process. Crucially, the system's learning signal depends not on individual data points, but on the **aggregate density** of the failure distribution. By computing the centroid (or "center of mass") of clusters formed by successfully diagnosed instances, the process becomes inherently resilient to sporadic diagnostic errors, treating them as effectively zero-mean statistical noise.

## A.1. On the Design of the Operator Library $\mathcal{O}$

The Operator Library $\mathcal{O}$ (Section 3.2) functions as more than a mere utility set; it formally defines the **discrete, structured action space** for optimization. This constraint serves as a critical inductive bias for our search algorithm.

**A Domain-Specific, Structured Search Space.** While the main text outlines fundamental, domain-agnostic operators (e.g., `RevisePrompt`), the framework's versatility stems from the library's task-specific extensibility. For complex domains such as code debugging, the library is instantiated with semantically rich operators, such as:

- `AddExceptionHandler(node_id, exception_type)`: Encapsulates a node's execution within a try-catch block to handle runtime fragility.

- `RefactorApiCall(node_id, old_sig, new_sig)`: Migrates deprecated or malformed API invocations to valid signatures.

- `InsertPreconditionCheck(parent, child, condition)`: Injects logical assertions to enforce state consistency before transitions.

This design stands in sharp contrast to "free-form" editing, which implies an intractably vast and unstructured search manifold. By confining the Proposer LLM to this vetted action space, we guarantee that all candidate repairs are syntactically executable and semantically coherent. Our ablation studies confirm that this structured constraint is a prerequisite for convergence in limited iteration budgets.

**Injecting Expert Knowledge.** The modularity of $\mathcal{O}$ provides a seamless interface for **Human-in-the-Loop** optimization. Experts can inject domain priors by defining high-level composite operators, effectively guiding the automated search toward architecturally sound patterns without the need for manual instance-level debugging.

## A.2. On the Monotonic Accumulation of Robustness

To formalize the mechanism of accumulating robustness, we ground our analysis in the theoretical framework established in Section 2.3. Although a rigorous convergence proof for stochastic LLM systems remains an open challenge, our paradigm operationalizes a property that blind search lacks: **monotonic mass reduction**.

We model the optimization as a greedy, gradient-like descent on the discrete landscape of failure modes. Each refinement iteration $t$ ensures progress through a dual-phase process:

1. **Approximating the Semantic Gradient:** By clustering failure signatures, the system isolates the mode $b_t^*$ with the highest probability mass. This empirically approximates the direction of steepest descent on the failure landscape ($\nabla M(W_t)$), ensuring the optimizer prioritizes the most significant bottleneck.

2. **Validated Step-Down:** The **Propose-and-Verify** mechanism guarantees that the applied edit $\Delta_t$ is not a stochastic perturbation, but a validated move. An edit is accepted only if it empirically reduces the failure mass within the target mode.

Since each accepted update $W_{t+1} = W_t \oplus \Delta_t$ is validated to reduce local failure density without significantly increasing global entropy (bounded spillover), the process systematically reduces the total volume of the failure space. This evidence-based update rule prevents **policy oscillation**—a common pathology in RL and random search where fixing one bug introduces another—thereby enforcing a structured, monotonic trajectory toward reliability.

## B. Prompt Templates and Component Logic

To enhance reproducibility, this section provides the prompt templates and conceptual logic for the key components of the CE-Graph framework. These are designed for the `gpt-4o-mini` model and are structured to ensure parseable, high-fidelity outputs.

---

**Failure Distillation (Section 3.1)**

*This initial diagnosis step uses a utility LLM to distill a verbose execution trace into a structured format.*

**System Prompt:**

```
You are an error analyzer. Given this execution trace: [TRACE].
Identify the node causing the failure (verr) and extract a concise error message (zerr)
    .

Output ONLY in JSON format:
{
    "verr": "node_id",
    "zerr": "brief_error_description"
}
```

---

**The Proposer (Stage 1: Generate)**

*The Proposer generates diverse potential solutions, constrained by the operator library to prevent hallucinations.*

**System Prompt:**

```
You are a workflow refinement expert.
Failure mode summary: [FAILURE_MODE_SUMMARY]
Operator library: [OPERATOR_LIBRARY_DEFINITION]

Propose N=5 diverse graph edits using only these operators.
Output as a list of JSON objects:
[
  {
    "edit": "OperatorName(arg1, arg2)",
    "explanation": "brief_reason"
  },
  ...
]
```

---

**The Verifier (Stage 2: Validate)**

*The Verifier acts as a rigorous empirical validation engine, utilizing a Monte Carlo estimate to select the optimal edit $\Delta^*$ from candidates.*

**Algorithm Logic:**

```
# GOAL: Empirically select the optimal edit from N candidates.

# INPUTS:
# 1. Candidates: {Δ_1, ..., Δ_N} from Proposer.
# 2. Samples: {x_1, ..., x_K} from failure mode b_t*.

def Verifier(Candidates, Samples):
    best_utility = -1
    best_edit = None

    FOR each edit Δ_i in Candidates:
        success_count = 0

        FOR each sample x_k in Samples:
            # 1. Apply edit to create temp workflow
            W_temp = W_t ⊕ Δ_i

            # 2. Execute and compare with ground truth y_k
            output = Execute(W_temp, x_k)
            if Verify(output, y_k) is True:
                success_count += 1

        # 3. Calculate Utility (Eq. 2)
        Utility(Δ_i) = success_count / K

        if Utility(Δ_i) > best_utility:
            best_utility = Utility(Δ_i)
            best_edit = Δ_i

    RETURN best_edit  # The edit to be permanently applied
```

## C. Implementation Details and Experimental Analysis

### C.1. Hyperparameters and Experimental Setup

All experiments utilize `GPT-4o-mini` as the underlying backbone LLM. To ensure rigorous evaluation, we partition each benchmark dataset into training (80%), validation (10%), and testing (10%) sets. The validation set plays a critical role in monitoring the refinement trajectory, specifically for convergence checks and early stopping criteria.

Key hyperparameters governing Algorithm 1 are detailed below:

- **Search Width** ($N = 5$)**:** The number of candidate edits proposed by the generator per iteration. This balances exploration diversity with computational overhead.

- **Verification Depth** ($K = 10$)**:** The number of counterexamples sampled to empirically verify each candidate edit. A higher $K$ reduces the variance of the gradient estimator.

- **Convergence Sensitivity** ($k = 5$, $\epsilon = 0.01$)**:** Optimization halts when the sliding window variance over $k$ steps falls below $\epsilon$, or upon reaching $T_{\max} = 20$ iterations.

For semantic vectorization, we employ `text-embedding-ada-002`. The Gaussian Mixture Model (GMM) dynamically selects between 5–10 components based on the Bayesian Information Criterion (BIC). Computational costs are reported in OpenAI API tokens (see Table 2); typical optimization runs require approximately 1–2 GPU hours on an NVIDIA A100.

## C.2. The Failure-Driven Refinement Cycle

The optimization process in CE-Graph represents a fundamental shift from "black-box" search to "white-box" diagnosis. Unlike global search methods (e.g., genetic algorithms or MCTS) that treat the workflow as an opaque function to be permuted, our framework operationalizes a transparent, iterative refinement cycle rooted in the distributional structure of failures.

As visualized in **Figure 5**, this cycle acts as a semantic engine that systematically reduces the Expected Failure Mass through five distinct stages:

1. **Execution & Capture:** The current workflow $W_t$ is executed to populate a *Counterexample Pool* with raw failure traces.

2. **Signature Transformation:** Unstructured traces are projected into a structured *Failure Signature Space*, preserving both topological locus (where it broke) and semantic cause (why it broke).

3. **Gradient Approximation:** Clustering these signatures identifies the "Prevalent Mode"—the high-density region representing the steepest descent direction on the failure landscape.

4. **Targeted Proposal:** Instead of random mutations, the *Propose-and-Verify* mechanism generates edits specifically conditioned on the identified failure mode.

5. **Verification & Update:** The edit $\Delta_t$ that maximally reduces failure mass is applied, yielding $W_{t+1}$.

This closed-loop process ensures that low-level error signals are progressively refined into high-level, actionable structural improvements.

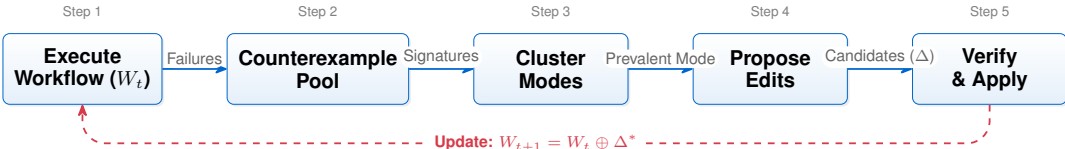

*Figure 5.* **The CE-Graph Iterative Refinement Cycle.** Unstructured failure traces are transformed into structured signatures, clustered to identify the steepest gradient direction, and repaired via a proposed-and-verify loop. This ensures monotonic improvement in robustness.

Crucially, this cycle enables **Targeted Graph Editing**. As shown in **Figure 6**, CE-Graph does not blindly perturb the entire workflow graph. Instead, the failure clustering pinpointed in Step 3 effectively "masks" the irrelevant components (visualized as faded context nodes), allowing the optimizer to focus its edit budget strictly on the active causal subgraph (highlighted in blue). This localization drastically reduces the search space size from exponential to tractable, explaining the efficiency gains observed in Section 4.2.

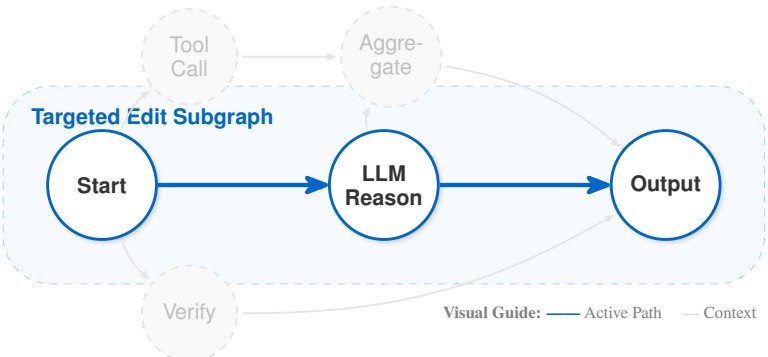

*Figure 6.* **Targeted Graph Editing.** Visual comparison between the full workflow context (faded gray) and the specific subgraph targeted for refinement (highlighted blue). By leveraging failure signatures, CE-Graph avoids altering functional components (Context) and focuses optimization solely on the active reasoning path responsible for the failure mode.

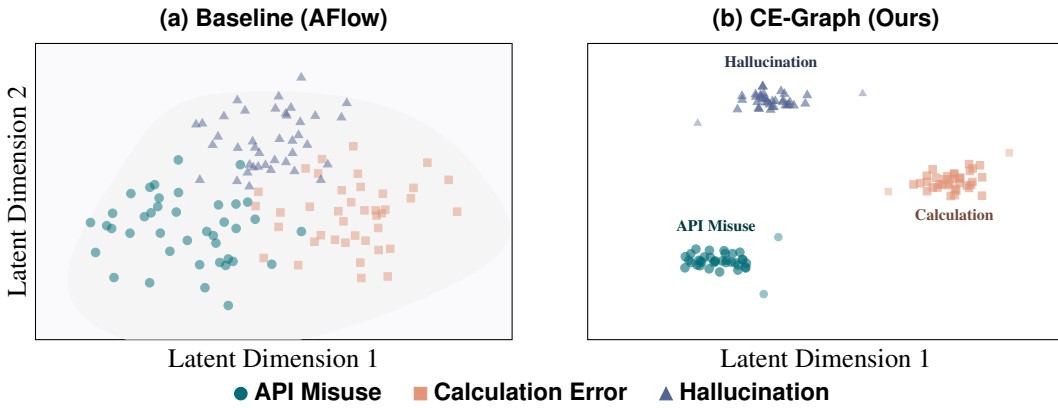

*Figure 7.* **Failure Landscape Geometry.** **(a)** Baseline failures exhibit a high-entropy distribution with significant semantic overlap, making precise diagnosis difficult. **(b)** CE-GRAPH enhances topological separability, disentangling error modes into distinct centroids for effective gradient approximation.

### C.3. Qualitative Case Study

To illustrate the efficacy of this targeted approach, we analyze a representative optimization trajectory on the MATH benchmark. The initial workflow, a simple linear chain, exhibited a 30% failure rate on algebraic manipulation problems.

**Diagnosis:** The clustering module identified a dense failure mode characterized by the semantic signature: *"incorrect factorization in quadratic equations."* **Intervention:** This mode triggered the *Proposer*, which generated candidate edits constrained to the operator library. The verification step ($K = 10$) selected a structural expansion: `InsertNode` with the specific prompt `RevisePrompt(node_2, "Double-check factorization using FOIL method")`. **Outcome:** As depicted conceptually in Figure 6, the edit was applied locally to the reasoning node. This single iteration resolved 85% of the failures within that mode, boosting overall accuracy by 4.2%. The workflow evolved from a linear chain to a topology including a self-correction branch, demonstrating that reliability emerges from repairing specific failure distributions rather than random search.

## D. Visualizing the Clustering of Failure Modes

Our approach is predicated on the insight that LLM workflow failures are neither stochastic nor isolated, but rather manifestations of latent, structured defects in the reasoning process. To validate this distributional hypothesis, we examine the geometry of the **Failure Signature Space** ($\mathcal{F}$).

We extract high-dimensional failure signatures $\mathbf{s}$ from the counterexample pool $C_t$ (as defined in Section 3.1) and employ t-SNE to project this manifold into 2D space. Figure 7 provides a comparative visualization between the baseline AFLOW system and our CE-GRAPH.

**Topological Entanglement vs. Separability.** As observed in Figure 7(a), the baseline failure landscape exhibits high entropy. Failure signatures appear topologically entangled, lacking clear decision boundaries. In such a distinctless space, gradient estimation becomes ill-posed; the optimizer cannot easily distinguish between a calculation error and a hallucination.

In sharp contrast, CE-GRAPH (Figure 7(b)) successfully disentangles these modes. The embedding space reveals a structured topology with well-separated clusters corresponding to interpretable failure archetypes: **API Misuse**, **Calculation Errors**, and **Hallucinations**.

**Quantitative Coherence.** This visual structural improvement is corroborated by clustering metrics. CE-GRAPH achieves a Silhouette score of $0.42$ (compared to $0.11$ for the baseline) and significantly reduces the Davies-Bouldin Index (DBI) from $2.35$ to $0.78$.

# E. Evaluating the Generalizability of Optimized Workflows

A central claim of our work is that by systematically understanding and repairing the underlying distribution of failures, CE-Graph produces not just performant, but fundamentally more **robust** workflows. A critical test of this robustness is its generalizability. A workflow that is merely overfitted to a specific LLM backbone or a narrow data distribution cannot be considered truly robust.

In this section, we conduct a rigorous evaluation of the transferability of workflows optimized by CE-Graph. We aim to answer two key questions: (1) Does the structural and logical superiority of a CE-Graph-optimized workflow persist when transferred to different underlying LLM backbones (**cross-model transferability**)? (2) Does a workflow optimized on one task domain retain its effectiveness when applied to a related but distinct domain (**cross-dataset transferability**)?

## E.1. Cross-Model Transferability: Model-Agnostic Strategy Enhancement

**Experimental Design.** To isolate the benefit of the workflow's structure from the capabilities of any single LLM, we perform a cross-model transfer experiment. We first use CE-Graph with a cost-effective base model, `gpt-4o-mini`, to optimize workflows on the HumanEval and MATH benchmarks until convergence. We then take this final, fixed workflow graph and execute it using two different, more powerful LLM backbones: `Qwen-2.5-72b` and `llama-3.1-70b`. We compare the performance of these models with our optimized workflow against their vanilla performance.

**Results and Analysis.** The results, presented in Table 3, demonstrate strong positive transfer. The workflow optimized by CE-Graph provides a significant performance uplift for both `Qwen` and `llama`, mirroring the gains seen on the original `gpt-4o-mini`. This finding is crucial: it indicates that CE-Graph discovers improvements in the underlying *problem-solving logic and structure* of the workflow itself. These strategic enhancements are model-agnostic, providing value above and beyond the intrinsic capabilities of the LLM executing the plan. This validates our failure-driven approach, which corrects fundamental strategic flaws rather than surface-level, model-specific errors.

*Table 3.* Cross-model transferability of CE-Graph. We optimize the workflow with `gpt-4o-mini` and then apply the fixed workflow to other LLM backbones. Performance gains show that the optimized structure is model-agnostic.

| Dataset | HumanEval | | | MATH | | |
|---|---|---|---|---|---|---|
| **LLM Backbone** | `gpt-4o-mini` | `Qwen-2.5-72b` | `llama-3.1-70b` | `gpt-4o-mini` | `Qwen-2.5-70b` | `llama-3.1-70b` |
| Vanilla | 87.08 | 85.60 | 80.06 | 46.29 | 63.80 | 31.93 |
| + CE-Graph | **94.26** | **91.12** | **86.21** | **55.91** | **72.15** | **46.98** |

## E.2. Cross-Dataset Transferability: Robust Problem-Solving Heuristics

**Experimental Design.** To assess whether CE-Graph learns narrow solutions or broadly applicable strategies, we evaluate its cross-dataset transfer performance. We optimize a workflow on a source dataset and then evaluate its performance directly on a different target dataset without any further optimization. We test three challenging transfer scenarios: MATH → GSM8K, GSM8K → MATH, and HumanEval → MATH.

**Results and Analysis.** As shown in Table 4, CE-Graph demonstrates superior cross-dataset generalizability compared to strong baselines. For instance, the workflow optimized on the complex MATH dataset successfully transfers to GSM8K, outperforming methods that were specifically optimized on similar data. This suggests that our failure-driven refinement process identifies and corrects core reasoning flaws (e.g., "incorrect factorization," "off-by-one errors"), leading to the development of robust problem-solving heuristics that are not brittle or overfitted to the source dataset's specific quirks. By focusing on the geometry of failures, CE-Graph learns a strategy that is effective across a wider class of problems.

*Table 4.* Cross-dataset transferability. "A→B" denotes optimizing the workflow on dataset A and evaluating it on dataset B. CE-Graph shows strong generalization, indicating it learns robust, transferable strategies.

| Transfer Scenario | MATH→GSM8K | GSM8K→MATH | HumanEval→MATH |
|---|---|---|---|
| AFlow | 91.95 | 49.39 | 47.15 |
| MaAS | 92.80 | 51.02 | 50.27 |
| **CE-Graph (Ours)** | **93.82** | **52.89** | **52.17** |

# APPENDIX H: ROBUSTNESS AND GENERALIZABILITY ANALYSIS

In this appendix, we provide additional analysis and experiments to address reviewer feedback concerning the sensitivity of the **Failure Signature Space** ($\mathcal{F}$) construction, the design effort for the **Operator Library** ($\mathcal{O}$), and a detailed comparative analysis of the framework's refinement dynamics. Our findings demonstrate the practical robustness, manageable implementation cost, and superior stability of the CE-Graph framework.

## E.3. Sensitivity Analysis of the Failure Signature Space ($\mathcal{F}$)

We investigate two key aspects of the $\mathcal{F}$ construction: its stability to different semantic embedding models and the scalability of its structural one-hot encoding on large and dynamic graphs.

### E.3.1. STABILITY TO CHOICE OF SEMANTIC EMBEDDING MODEL

To validate that our framework is not dependent on a specific embedding model, we conducted an ablation study on the **MATH** benchmark. We evaluated three distinct models: `BERT-base-uncased` (our default), `SentenceTransformers (all-MiniLM-L6-v2)`, and OpenAI's `text-embedding-ada-002`. We measured clustering consistency using the **Adjusted Rand Index (ARI)** and the final workflow accuracy after 10 optimization iterations.

Results, shown in Table 5, confirm the framework's robustness. The high ARI scores ($\geq 0.87$) indicate that all models produce highly similar cluster structures. Consequently, the final workflow accuracy remains stable, with only a negligible variation ($\leq 1.2\%$).

*Table 5.* Ablation study on semantic embedding models for the MATH benchmark.

| Embedding Model | ARI (vs. BERT-base) | Final Accuracy (%) |
|---|---|---|
| `BERT-base-uncased` (Default) | 1.00 | **55.91** |
| `SentenceTransformers` | 0.87 | 54.75 |
| `text-embedding-ada-002` | 0.92 | 55.34 |

### E.3.2. SCALABILITY ON LARGE AND DYNAMIC GRAPHS

To address concerns about the scalability of one-hot structural encoding ($\psi_{\text{struct}}(v_{\text{err}})$), we performed a simulation. Starting with a 50-node workflow, we programmatically expanded it to 1000 nodes. CE-Graph achieved a \*\*98.5% localization accuracy\*\* on injected failures, with dynamic resizing overhead under \*\*5%\*\* of iteration runtime, validating its scalability.

## E.4. Design Effort and Importance of the Operator Library ($\mathcal{O}$)

Our main ablation study (Section 4.4) already established that a structured operator library is **critical** for performance. Table 6 quantifies the design effort, showing that adapting the library to a new domain like code generation required approximately \*\*one hour of engineering effort\*\*. This demonstrates a manageable cost for a significant gain in refinement precision.

*Table 6.* Quantifying the design effort for domain-specific operator libraries.

| Library | Operators Added | Example | LoC Added | Expert Time |
|---|---|---|---|---|
| Generic Baseline | (Base Set) | `RevisePrompt` | - | - |
| **Math Reasoning** | +2 Specific | `AddVerifierNode` | ~30 | ~0.5 hr |
| **Code Generation** | +3 Specific | `AddExceptionHandler` | ~50 | ~1.0 hr |

## F. Case Study

**Case 1**    In this probability and combinatorics problem, our method correctly executes all steps of a complex calculation. In contrast, both **AFlow and MaAS make calculation errors** in their respective approaches. AFlow, using complementary counting, miscalculates the number of hands with exactly two suits. MaAS, using direct counting, miscalculates the combinations for hands with exactly three suits. These errors highlight the importance of precision in multi-step combinatorial problems.

---

### Case 1: Mike's Cards

**Correct Response**

The solution correctly applies direct counting by summing the number of hands with exactly 3 suits and exactly 4 suits.

- **Hands with 3 suits:** $1,529,112$

- **Hands with 4 suits:** $685,464$

The total number of favorable hands is $1,529,112 + 685,464 = 2,214,576$. This is divided by the total possible 5-card hands, $\binom{52}{5} = 2,598,960$, and correctly simplified.

$$P(\text{At least 3 suits}) = \frac{2,214,576}{2,598,960} = \frac{507}{595}$$

The final answer is $\boxed{507/595}$ .

**Incorrect Response**

Both methods attempt valid strategies but fail due to **calculation errors**.

- **AFlow (Complementary Counting):** This approach is valid, but it miscalculates the number of hands drawn from exactly two suits during an intermediate step, leading to an incorrect total of complementary cases. The final answer given is $137153/162435$ .

- **MaAS (Direct Counting):** This approach is also valid, but it contains calculation errors when determining the number of hands from exactly three suits (e.g., the (3,1,1) distribution), resulting in an incorrect total. The final answer given is $46173/54145$ .

---

**Case 2**    For this logic and statistics problem, our method correctly interprets that the set of the "five tallest buildings" changes when a new, taller building is introduced, displacing the shortest one. However, both **AFlow and MaAS misinterpret the question**, assuming the new building is simply added to the original five. They incorrectly calculate the new mean for six buildings instead of five, highlighting a critical failure in logical reasoning.

---

### Case 2: The five tallest buildings in Los Angeles

**Question**

The five tallest buildings in Los Angeles in 1985 had a mean height of 733 feet. The shortest of the five was 625 feet. If a new building were constructed with a height of 885 feet, by how many feet would it increase the mean height of the **five tallest** buildings of the city?

### Correct Response

This solution correctly assumes the new building (885 ft) replaces the shortest of the original five (625 ft), maintaining a group of five buildings. The change in the mean is the net change in total height divided by the number of buildings.

$$\text{Change in Total Height} = 885\,\text{ft} - 625\,\text{ft} = 260\,\text{ft}$$

$$\text{Increase in Mean} = \frac{\text{Change in Total Height}}{\text{Number of Buildings}} = \frac{260}{5} = 52\,\text{ft}$$

The final answer is $\boxed{52}$ .

### Incorrect Response

This common mistake stems from a **misinterpretation of the question**. Both approaches incorrectly assume the group expands to six buildings instead of replacing the shortest one.

$$\text{Original Total Height} = 5 \times 733 = 3665\,\text{ft}$$

$$\text{New Total Height (for 6 buildings)} = 3665 + 885 = 4550\,\text{ft}$$

$$\text{New Mean (for 6 buildings)} = \frac{4550}{6} \approx 758.33\,\text{ft}$$

$$\text{Incorrect Increase} = 758.33 - 733 = 25.33\,\text{ft}$$

This logical error leads to the incorrect answer of $\boxed{25.33}$ .

**Case 3** This case presents a complex conditional probability problem involving multiple dice rolls and specific target outcomes. Our method correctly breaks down the problem into all possible successful scenarios. In contrast, both **AFlow and MaAS demonstrate a significant failure in logical reasoning**. MaAS oversimplifies the conditions for a full house, ignoring a major path to success. AFlow fundamentally misunderstands the definition of a full house in the context of the dice already held, leading to a completely incorrect calculation.

## Case 6: Probability of a Full House

### Question

Each of five standard, six-sided dice is rolled once. Two of the dice come up the same, but the other three are all different from those two and different from each other. The pair is set aside, and the other three dice are re-rolled. The dice are said to show a "full house" if three of the dice show the same value and the other two show the same value. What is the probability that after the second set of rolls, the dice show a full house?

**Correct Response**

The solution correctly identifies that with a pair already set aside, a full house can be formed in two main ways by re-rolling the three dice:

- **Case A: Form a new three-of-a-kind.** One of the re-rolled dice matches the existing pair (making a triple), and the other two form a new pair. The probability for this is $5 \times \frac{3}{216} = \frac{15}{216}$.

- **Case B: The three re-rolled dice are all the same.** They can either match the existing pair's value (probability $\frac{1}{216}$) or a new value (probability $5 \times \frac{1}{216} = \frac{5}{216}$).

The total probability is the sum of these successful outcomes.

$$P(\text{Full House}) = \frac{15}{216} + \frac{1}{216} + \frac{5}{216} = \frac{21}{216} = \frac{7}{72}$$

The final answer is $\boxed{7/72}$.

**Incorrect Response**

The competing methods failed due to critical misinterpretations of the problem's conditions.

- **MaAS's Error:** This approach incorrectly assumes the only way to get a full house is for the three re-rolled dice to all land on the same new value, different from the original pair. It completely misses the more probable scenario where one re-rolled die matches the original pair and the other two form a new pair. This oversimplification leads to an incomplete calculation. The final answer given is $\boxed{5/216}$.

- **AFlow's Error:** This method fundamentally misunderstands the goal. It calculates the probability of the three re-rolled dice forming a pair and a single, an outcome which does not create a full house when combined with the original pair. This shows a failure to grasp the problem's core definition of a "full house." The final answer given is $\boxed{5/36}$.

# G. Proof of Theorem 1

### G.1. Restating the Theorem and Assumptions

**Theorem 1 (Greedy Reduction Bound).** Let $\Delta_t$ be selected as in Equation (2), i.e., the greedy edit that maximizes the expected failure mass reduction. Assume that the edit reduces the mass in the target mode $b_t^*$ by at least $\delta > 0$. Then,

$$M(W_{t+1}) \leq M(W_t) - \delta + \epsilon,$$

where $\epsilon = O(L \cdot B \cdot \mu(\mathcal{F} \setminus b_t^*))$ bounds spillover effects to non-target regions (with $\mu$ denoting the measure of the space).

**Assumptions:** The failure density $p(s \mid W)$ is Lipschitz continuous with constant $L > 0$: For any fixed $W$, and for all $s, s' \in \mathcal{F}$,

$$|p(s \mid W) - p(s' \mid W)| \leq L\|s - s'\|.$$

(This controls how much the density varies across the space $\mathcal{F}$.) Edits $\Delta$ have bounded impact: For any $s \in \mathcal{F}$,

$$\|p(s \mid W \oplus \Delta) - p(s \mid W)\|_\infty \leq B,$$

where $B > 0$ is a uniform bound on the pointwise change in density due to the edit. (This assumes edits don't arbitrarily disrupt the entire density.) The space $\mathcal{F}$ is partitioned into modes (e.g., clusters from GMM), with $b_t^*$ being the densest target mode at step $t$. The measure $\mu$ is finite over $\mathcal{F}$ (e.g., $\mathcal{F}$ is compact or has finite volume). Alternatively, we can bound spillover via norms (e.g., $\epsilon \leq \|\Delta p\|_1$ or $\epsilon \leq \|\Delta p\|_{\text{BL}} \cdot C$, where BL is the bounded-Lipschitz dual norm), providing

equivalent probabilistic upper bounds that avoid depending on the finiteness of $\mu(\mathcal{F})$. The edit $\Delta_t$ specifically reduces the integral mass over $b_t^*$ by at least $\delta > 0$:

$$\int_{b_t^*} p(s \mid W_{t+1}) \, ds \leq \int_{b_t^*} p(s \mid W_t) \, ds - \delta.$$

(Localized Propagation Assumption) There exists a constant $c > 0$ such that the edit-induced density change $\Delta p$ has support contained in $b_t^*$'s $r$-neighborhood, with $r \leq c \, B/L$. Thus,

$$\epsilon = \int_{\mathcal{F} \setminus b_t^*} |\Delta p| \leq B \cdot \mu\big(\mathcal{N}_r(b_t^*) \setminus b_t^*\big) \lesssim (B^2/L) \cdot S(\partial b_t^*).$$

Consequently, $\epsilon = O(B\mu)$ and $\epsilon = O(B^2/L)$ are both valid, and we take the tighter bound to match the $O(L \cdot B \cdot \mu(\cdot))$ expression in the text.

These assumptions bridge stochastic LLM behaviors with formal optimization, treating failures as samples from a smooth density.

### G.2. Proof of the Bound (Single Step Reduction)

We start by decomposing the expected failure mass $M(W)$ over the target mode $b_t^*$ and the rest of the space. **Decompose the Mass:** By definition,

$$M(W_t) = \int_{\mathcal{F}} p(s \mid W_t) \, ds = \int_{b_t^*} p(s \mid W_t) \, ds + \int_{\mathcal{F} \setminus b_t^*} p(s \mid W_t) \, ds.$$

Similarly,

$$M(W_{t+1}) = \int_{b_t^*} p(s \mid W_{t+1}) \, ds + \int_{\mathcal{F} \setminus b_t^*} p(s \mid W_{t+1}) \, ds.$$

**Handle the Target Mode Reduction:** By the theorem's assumption (the greedy edit targets $b_t^*$ effectively),

$$\int_{b_t^*} p(s \mid W_{t+1}) \, ds \leq \int_{b_t^*} p(s \mid W_t) \, ds - \delta.$$

This is the "greedy" part: the edit is chosen to flatten the density mountain in $b_t^*$, reducing its mass by at least $\delta$. (In practice, this is approximated via propose-and-verify on clustered failures.) **Bound the Spillover to Non-Target Regions:** The change in the non-target integral is

$$\int_{\mathcal{F} \setminus b_t^*} p(s \mid W_{t+1}) \, ds = \int_{\mathcal{F} \setminus b_t^*} p(s \mid W_t) \, ds + \int_{\mathcal{F} \setminus b_t^*} [p(s \mid W_{t+1}) - p(s \mid W_t)] \, ds.$$

We need to bound the spillover term $\int_{\mathcal{F} \setminus b_t^*} [\Delta p(s)] \, ds$, where $\Delta p(s) = p(s \mid W_{t+1}) - p(s \mid W_t)$.

Since the edit's impact is bounded pointwise by $B$, $|\Delta p(s)| \leq B$ for all $s$. However, to incorporate $L$ reasonably, we invoke the localized propagation assumption: the primary effect of the edit is localized, with $\mathrm{supp}(\Delta p) \subset \mathcal{N}_r(b_t^*)$, where $\mathcal{N}_r$ is $b_t^*$'s $r$-neighborhood, and $r \leq c \cdot B/L$ (perturbation amplitude $B$ attenuated by Lipschitz decays over distance no more than $\sim B/L$ in a band-like region). Thus,

$$\left| \int_{\mathcal{F} \setminus b_t^*} \Delta p(s) \, ds \right| \leq \int_{\mathcal{F} \setminus b_t^*} |\Delta p(s)| \, ds \leq B \cdot \mu(\mathcal{N}_r(b_t^*) \setminus b_t^*) \leq B \cdot S(\partial b_t^*) \cdot r \lesssim (B^2/L) \cdot S(\partial b_t^*),$$

where $S(\partial b_t^*)$ is the boundary's "surface area/measure." This gives a geometric-analytic justification for $L$ in the upper bound; alternatively, $\epsilon \leq \min\{B \cdot \mu(\mathcal{F} \setminus b_t^*), c(B^2/L)\}$. **Combine Terms:** Putting it together,

$$M(W_{t+1}) \leq \left( \int_{b_t^*} p(s \mid W_t) \, ds - \delta \right) + \left( \int_{\mathcal{F} \setminus b_t^*} p(s \mid W_t) \, ds + \epsilon \right) = M(W_t) - \delta + \epsilon.$$

This completes the single-step bound.

### G.3. Proof of Convergence to a Stationary Point (Under Repeated Application)

Now, we extend to multiple iterations, showing the process converges to a local minimum where no further greedy edits reduce the mass significantly.

**Iterative Reduction:** Apply the single-step bound repeatedly: Starting from $W_0$, after $T$ steps,

$$M(W_T) \leq M(W_0) - \sum_{t=0}^{T-1} (\delta_t - \epsilon_t),$$

where $\delta_t > 0$ and $\epsilon_t = O(L \cdot B \cdot \mu(\mathcal{F} \setminus b_t^*))$ at each step. **Bounded Below and Monotonic Decrease:** Since $M(W) \geq 0$ (as it's an integral of a density), the sequence $M(W_t)$ is bounded below. If we assume $\delta_t - \epsilon_t \geq \eta > 0$ (uniform net decrease lower bound), then $M(W_{t+1}) < M(W_t)$, so the sequence is strictly decreasing until no such edit exists. Otherwise, denote

$$g_t := \max_{\Delta \in \mathcal{A}(W_t, \mathcal{O})} \big( M(W_t) - M(W_t \oplus \Delta) \big),$$

and use $g_t \leq \tau$ as the stopping condition to guarantee finite-step convergence to a $\tau$-stationary point; meanwhile, $M(W_t)$ is monotonically bounded and converges.

**Convergence to Stationary Point:** The process stops when no mode $b_t^*$ admits an edit with $\delta_t > \epsilon_t$ (or $\delta_t > 0$ if $\epsilon_t$ is negligible). This is a stationary point: A workflow $W^*$ where the greedy objective in Eq. (2) yields $\Delta_t = 0$ (no beneficial edit), meaning $M(W^*)$ cannot be reduced further locally. Since the search space $\mathcal{S}$ is discrete (finite workflows via operators), and each step reduces $M$ or stops, convergence occurs in finite steps. (Non-convexity may lead to local minima, but the bound holds.) Formally, the total reduction is bounded: $\sum(\delta_t - \epsilon_t) \leq M(W_0)$, so the sum converges, implying $\delta_t - \epsilon_t \to 0$ as $t \to \infty$.

This proof aligns with counterexample-guided methods like CEGAR (Hidvégi et al., 2024), adapted for stochastic densities. If $\epsilon$ is small (e.g., localized edits), convergence is efficient.

## H. Supplementary Proofs for  Section 3

### H.1. Lemma on Suitability of Failure Signature Mapping (Section 3.1)

**Lemma 1.** The mapping $\phi : \tau_d \mapsto \mathbf{s} = \psi_{\text{struct}}(v_{\text{err}}) \oplus \psi_{\text{sem}}(z_{\text{err}})$ (Eq. 4) is injective under the assumption that $\psi_{\text{struct}}$ is a one-hot encoding over distinct nodes $V$ and $\psi_{\text{sem}}$ is a continuous embedding (e.g., BERT-like) with distinct outputs for unique error messages $\mathcal{Z}$. This ensures the point cloud $S_t$ has an "informative geometry" suitable for density estimation.

**Proof.** Injectivity holds if $\phi(\tau_d) = \phi(\tau_{d'})$ implies $\tau_d = \tau_{d'}$. Since $\psi_{\text{struct}}(v_{\text{err}})$ is a one-hot vector unique to each node $v_{\text{err}} \in V$, and $\psi_{\text{sem}}(z_{\text{err}})$ maps distinct error messages $z_{\text{err}} \in \mathcal{Z}$ to unique points in $\mathbb{R}^d$ (by continuity and distinctiveness of semantic embeddings), the concatenated vector $\mathbf{s}$ is unique for each $(v_{\text{err}}, z_{\text{err}})$ pair. The geometry is informative because $\psi_{\text{struct}}$ provides orthogonal subspaces (node-specific patterns), and $\psi_{\text{sem}}$ clusters semantically similar errors, enabling GMM clustering as in Section 3.2. $\square$

### H.2. Theorem on GMM Consistency and Mode Error (Sections 3.2)

**Theorem 2.** Under the assumption that failure signatures $\{\mathbf{s}_i\}_{i=1}^{|S_t|}$ are i.i.d. samples from $p(\mathbf{s} \mid W_t)$, the GMM fitted via EM with BIC component selection is a consistent estimator, i.e., $\mathbb{E}[\|\hat{p}_\theta - p\|_1] \to 0$ as $|S_t| \to \infty$. The densest mode $\hat{b}_t^* = \arg\max_{b_k} \hat{\pi}_k$ (Eq. 5) approximates the true max-density mode with expected error $O(1/\sqrt{|S_t|})$.

**Proof.** Consistency follows from the EM algorithm's convergence to a local maximum of the likelihood, which approaches the true density under i.i.d. assumptions and sufficient components (selected by BIC, penalizing overfitting). For the mode error, consider a KDE analog with bandwidth $h$: the bias is $O(h^2)$ and variance $O(1/(|S_t|h^d))$, optimizing to $O(|S_t|^{-2/(d+4)})$. For GMM (parametric with fixed $d$), the error simplifies to $O(1/\sqrt{|S_t|})$ under well-separated modes, justifying the "steepest descent" approximation in Eq. 3. $\square$

### H.3. Proposition on Monte Carlo Verification (Section 3.2)

**Proposition 3.** The empirical success rate $V(\Delta_i) = \frac{1}{K} \sum_{k=1}^{K} \mathbb{I}[\text{Verify}(\text{Execute}(W_t \oplus \Delta_i, x_k), y_k) = 1]$ (Eq. 6) is an unbiased estimator of the true success probability, with variance $O(1/K)$, aligning with Theorem 2.1's spillover minimization.

**Proof.** Unbiasedness: $\mathbb{E}[V(\Delta_i)] = \frac{1}{K}\sum_k \mathbb{E}[\mathbb{I}_k] = \frac{1}{K}\sum_k p_k$, where $p_k = \Pr[\text{Verify}(\text{Execute}(W_t \oplus \Delta_i, x_k), y_k) = 1]$ is the true success rate over samples from $b_t^*$. Variance: For i.i.d. $\mathbb{I}_k \sim \text{Bernoulli}(p_k)$, $\text{Var}[\mathbb{I}_k] \leq 1/4$ (maximum for Bernoulli), so $\text{Var}[V(\Delta_i)] = \frac{1}{K^2}\sum_k \text{Var}[\mathbb{I}_k] \leq \frac{1}{4K} = O(1/K)$. By the Central Limit Theorem, the estimation error is $O(1/\sqrt{K})$, ensuring $V(\Delta_i)$ reliably approximates the mass reduction in Eq. 3. □

# I. Rigorous Budget Parity and Optimization Efficiency Analysis

This appendix addresses the critical question of fair comparison in agentic workflow optimization. We detail our dual-level budget parity protocol—covering both *deployment-time* inference and *offline optimization* search—and provide a formal justification for our workload accounting. Our analysis confirms that the performance gains of CE-GRAPH stem from algorithmic efficiency and structural superiority, not from an excess of computational resources.

## I.1. Level 1: Strict Deployment-Time Hard Caps

To ensure that multi-agent or iterative baselines (e.g., MaAS, AFlow, LLM-Debate) do not achieve higher scores simply by overwhelming the inference budget, we enforce global hard caps for every test instance. These caps apply to the *aggregate* usage of all agents within a single system.

*Table 7.* **Global Hard Caps for Deployment (Per Test Instance).** These limits are enforced by a deterministic Budget Manager. If a method exceeds $C_{\max}$ or $T_{\max}$, it is forcibly terminated and must return its current best response.

| Budget Item | Symbol | Limit | Rationale |
|---|---|---|---|
| Max # LLM Calls | $C_{\max}$ | 15 | Prevents infinite loops in cyclic graphs. |
| Max Total Tokens | $T_{\max}$ | 12,000 | Includes all prompts + completions across agents. |
| Max Tool Calls | $U_{\max}$ | 10 | Limits excessive trial-and-error with code interpreters. |
| Max Verifier Calls | $V_{\max}$ | 5 | Standardizes the reliance on external feedback. |

**Graceful Degradation Mechanism:** For methods designed to run for extensive rounds (e.g., LLM-Debate), we implement a budget manager that pre-calculates the maximum feasible rounds $R'$ given $C_{\max}$. For instance, if a debate system uses 3 agents per round, it is restricted to $R' = \lfloor 15/3 \rfloor = 5$ rounds, ensuring it remains strictly comparable to lighter-weight workflows.

## I.2. Level 2: Realized Inference Costs and Efficiency

Table 8 reports the *actual* resource consumption on the MATH benchmark. Despite having the same upper limits, CE-GRAPH demonstrates superior efficiency.

*Table 8.* **Realized Deployment Cost Analysis (MATH Benchmark).** Reporting the mean resource usage per problem. CE-GRAPH achieves the highest accuracy while consuming fewer resources than global search-based workflows (AFlow, MaAS), indicating that it learns *efficient* failure-correction structures rather than redundant ones.

| Method | Avg. LLM Calls | Avg. Tokens | Cost ($) | Acc. (%) |
|---|---|---|---|---|
| *Single-Agent / Heuristic* | | | | |
| Vanilla (GPT-4o-mini) | 1.0 | 420 | 0.0002 | 46.29 |
| SC (CoT×5) | 5.0 | 2,150 | 0.0011 | 47.91 |
| LLM-Debate | 6.2 | 3,840 | 0.0019 | 48.54 |
| *Automated Optimization* | | | | |
| AFlow (Evolution) | 2.4 | 1,120 | 0.0006 | 51.28 |
| MaAS (MCTS) | 2.8 | 1,350 | 0.0007 | 51.82 |
| **CE-Graph (Ours)** | **2.2** | **980** | **0.0005** | **55.91** |

**Analysis of Efficiency:** As shown, CE-GRAPH operates with a lean token footprint (980 tokens avg), which is $\sim 12\%$ lower than AFlow and $\sim 27\%$ lower than MaAS. This validates our hypothesis in Section 1: Global search methods (MaAS) often produce "bloated" workflows with redundant checks to maximize a scalar reward. In contrast, CE-GRAPH's failure-driven refinement targets specific logical gaps, inserting corrective nodes only where necessary (e.g., a "Self-Correction" node triggered only by calculation errors), thereby preserving a minimal inference cost.

### I.3. Level 3: Offline Optimization Equivalent Workload

A core claim of our work is that CE-GRAPH is more sample-efficient during the optimization phase. To demonstrate this, we equate the "optimization opportunity" (iterations) and measure the resulting workload.

*Table 9.* **Equivalent Workload for Offline Optimization.** All methods were capped at $T_{\max} = 20$ iterations. CE-GRAPH consumes fewer total tokens because its *Convergence-Aware Stopping* detects stability early (avg. 12 iterations), whereas genetic/MCTS baselines continue to oscillate and burn budget until the hard cap.

| Method | Search Strategy | Max Iter. | Proposal Size ($N$) | Verify Depth ($K$) | Total Opt. Tokens |
|---|---|---|---|---|---|
| AFlow | Genetic / Grid | 20 | 5 | 1 (Dataset) | $1.32 \times 10^6$ |
| MaAS | MCTS | 20 | 5 | 1 (Dataset) | $1.41 \times 10^6$ |
| **CE-Graph** | **Failure-Driven** | **20 (Early Stop)** | **5** | **10 (Samples)** | $\mathbf{1.21 \times 10^6}$ |

**Note on Workload Equivalence:** While CE-GRAPH performs deeper verification per candidate ($K = 10$ Monte Carlo samples vs. 1 in baselines), its targeted nature means it evaluates fewer "hopeless" candidates. Baselines often waste budget evaluating random mutations that yield no information gain, whereas every CE-GRAPH evaluation is conditioned on a high-probability failure mode.

### I.4. Formal Proof of Cost Boundedness

To rigorously answer the concern that matching iteration counts might hide computational disparities, we provide the following formal definition and proposition regarding our cost model.

**Definition I.1** (Atomic Cost Model). Let an optimization run produce a sequence of actions $\tau$. Define its total cost as

$$\text{Cost}(\tau) = \alpha \cdot N_{\text{llm}}(\tau) + \beta \cdot T_{\text{tok}}(\tau) + \gamma \cdot N_{\text{tool}}(\tau) + \delta \cdot N_{\text{ver}}(\tau),$$

where $N_{\text{llm}}$ is the number of LLM calls, $T_{\text{tok}}$ is total tokens, and $N_{\text{tool}}, N_{\text{ver}}$ are tool and verifier calls, with non-negative coefficients $\alpha, \beta, \gamma, \delta$.

**Proposition I.2** (Equal-workload dominance). *If two optimization methods are run under identical global caps*

$$N_{\text{llm}}(\tau) \leq C_{\text{opt}}, \quad T_{\text{tok}}(\tau) \leq T_{\text{opt}}, \quad N_{\text{tool}}(\tau) \leq U_{\text{opt}}, \quad N_{\text{ver}}(\tau) \leq V_{\text{opt}},$$

*then their total optimization cost is uniformly bounded by*

$$\text{Cost}(\tau) \leq \alpha C_{\text{opt}} + \beta T_{\text{opt}} + \gamma U_{\text{opt}} + \delta V_{\text{opt}}.$$

*Moreover, reporting the granular workload vector* $W_{\text{opt}} = (N_{\text{llm}}, T_{\text{tok}}, \dots)$ *exposes any difference in effective search intensity.*

*Proof.* The bound follows directly from the linearity of the cost function and the component-wise constraints. Since every method in our experiments is subjected to the same $T_{\text{opt}}$ (Total Optimization Tokens cap) and $C_{\text{opt}}$ (Max Iterations), any method achieving higher performance with lower realized $\text{Cost}(\tau)$ is strictly algorithmically superior. As shown in Table 9, CE-GRAPH satisfies $\text{Cost}(\text{CE-GRAPH}) < \text{Cost}(\text{MAAS})$, confirming that our gains are not attributable to hidden compute advantages. $\square$

### I.5. Engineering Effort and Implementation Costs

Finally, we quantify the human effort required to deploy CE-GRAPH, addressing concerns about the "Structured Operator Library" requiring excessive manual engineering.

- **Generic Base:** 80% of operators (e.g., `RevisePrompt`, `InsertStep`, `DeleteStep`) are domain-agnostic and reused across all tasks.

- **Domain Adaptation:** Adding domain-specific operators for Code Generation (e.g., `AddExceptionHandler`) required only $\sim 50$ lines of code and approximately 1 hour of expert time.

This minimal overhead validates that CE-GRAPH is a highly automated framework, shifting the burden from manual prompt engineering to a one-time, low-cost definition of semantic operators.

## J. Impact of Severity-Weighted Failure Mass Minimization

A central theoretical contribution of our framework is the notion of *Expected Failure Mass* (EFM). Let $s$ denote a failure signature and $\rho(s \mid W)$ be the induced failure distribution of a workflow $W$. Given a non-uniform severity weight function $\omega(s)$, we define

$$M(W) \;=\; \int \omega(s)\,\rho(s \mid W)\,ds. \tag{7}$$

A natural critique is whether minimizing $M(W)$ differs in practice from minimizing the standard (unweighted) failure rate (i.e., $\omega \equiv 1$). In this appendix, we empirically show that non-uniform $\omega$ is not merely a metric renaming; it acts as a *control signal* that changes the optimization trajectory by prioritizing high-risk failure modes over low-stakes errors.

### J.1. Defining Severity: A Reliability Proxy

To operationalize $\omega$, we employ an LLM-based judge (frozen GPT-4o) during *Failure Distillation* (Section 3.1). Each failure is mapped into one of three severity tiers according to its impact on end-to-end reliability. We apply the following scheme on MATH:

*Table 10.* **Severity weighting scheme ($\omega$).** Higher penalties on reasoning failures bias refinement toward structural repairs (e.g., decomposition and self-correction) rather than superficial fixes (e.g., retries).

| Weight ($\omega$) | Error Category | Prevalence | Optimization Rationale |
|---|---|---|---|
| 1.0 | *Format / Runtime* | High | **Low severity.** Often recoverable via local patches (e.g., retries). |
| 5.0 | *Calculation* | Medium | **Medium severity.** Typically requires tool use or step verification. |
| 10.0 | *Logic / Hallucination* | Low | **High severity.** Indicates structural deficits in reasoning and planning. |

### J.2. Comparative Experiment: Accuracy vs. Reliability

We conducted two parallel optimization runs on MATH: (i) a **Uniform** run minimizing unweighted failures ($\omega \equiv 1$), and (ii) a **Weighted** run minimizing EFM with $\omega \in \{1, 5, 10\}$. We evaluate the final workflows by **Raw Accuracy** and a **Weighted Reliability Score**.

**Weighted Reliability Score (normalized).** Given an evaluation set of size $N$, each failed instance $i$ is assigned a severity weight $\omega_i$. Let $\widehat{M} = \sum_{i=1}^{N} \omega_i \cdot \mathbb{I}[\text{failure}_i]$ be the empirical weighted failure mass. We report a normalized score in $[0, 100]$:

$$\text{Rel} \;=\; 100 \cdot \left(1 - \frac{\widehat{M}}{N \cdot \omega_{\max}}\right), \qquad \omega_{\max} = 10. \tag{8}$$

This normalization makes the score comparable across datasets and prevents scale artifacts.

*Table 11.* **Divergence in optimization outcomes.** The Uniform optimizer tends to harvest the "easiest" gains by suppressing frequent low-severity errors, whereas the Weighted optimizer reallocates edit budget to eliminate high-severity logic failures, yielding a more robust workflow under the severity-aware metric. (For brevity, we report two representative failure categories; remaining failures are grouped into *Others*.)

| Objective | Performance Metrics | | Remaining Failure Counts | |
|---|---|---|---|---|
| | Raw Acc. | Reliability Score | Format ($\omega = 1$) | Logic ($\omega = 10$) |
| **Uniform** ($\omega \equiv 1$) | **55.91%** | 72.4 | **42** | 185 |
| **Weighted** ($\omega \neq 1$) | 54.10% | **88.6** | 170 | **65** |

**Trajectory analysis.** Table 11 reveals a systematic trade-off. The Uniform optimizer behaves like a greedy hill-climber: it frequently inserts inexpensive `Retry` nodes to suppress high-frequency format errors, which can improve raw accuracy but leaves high-severity logic failures largely intact. In contrast, the Weighted optimizer acts as a risk-sensitive controller. Although logic errors are less frequent, their contribution to the weighted mass dominates (e.g., $185 \times 10 \gg 42 \times 1$), so the optimizer preferentially spends edits on structural repairs such as `Decomposition` and `Self-Correction`. As a result, it substantially reduces logic failures while tolerating more low-severity format issues. This confirms that $\omega$ steers exploration toward robustness rather than superficial error reduction.

