# OpenReview forum: "Failure-Driven Workflow Refinement"
_ICML.cc/2026/Conference — ICML 2026 spotlight_

### Official Review · Reviewer_dGFx · 2026-02-27

**Soundness:** 4
**Presentation:** 3
**Significance:** 4
**Originality:** 4
**Overall Recommendation:** 5
**Confidence:** 4

**Summary:**

The authors introduce an approach on modeling the failure topology in workflows executed by LLM agents. This is different from the outcome-based approach that has been used so far where only the outcome for each executed workflow is seen without trying to understand why and where the execution of the workflow has failed.

**Compliance With Llm Reviewing Policy:**

Affirmed.

**Final Justification:**

The authors addressed all my comments and I kept my score

**Key Questions For Authors:**

- How do you ensure that the one-hot encoding produced from the graph’s node position is consistent for different workflow graphs and that it captures enough information about the location of the node?
- You are concatenating dense with sparse embeddings which is used as input to the GMM. How does this affect the modeling of the failure space?
- I am assuming branches are due to if/else? What happens when a workflow involves a loop? A bit more details on the structure of this workflow graph would be welcome.

**Limitations:**

yes

**Strengths And Weaknesses:**

## Strengths:
- Modeling the failure distribution of the workflows to have an interpretable way to diagnose why the workflow failed. This is different from the outcome-based evaluation method we have today where we only observe the end result on whether the workflow succeeded without knowing why it failed.
- Application of GMMs for this use-case is perfect as you model the failure landscape and their severity. I think the combination of GMM along with proposing graph edits with LLMs to minimize expected failure risk is elegant.

## Weaknesses:
- How do you know that during the failure distillation that the LLM will narrow down to the correct error-producing node? In subtle errors this may be challenging. In such cases mistake will cascade and will affect the construction of the signature space F.
- As per my understanding embedding model for $ψ_{sem}$ is pre-trained. How much would poor embeddings produced affect the accurate representation of failure signature space F? Would it help if we gather positive/negative labels of pairs of error messages and train with contrastive learning a better embedding model?

Small typos/references:
- undefinedwiechowski et al., 2022
- Figure “I have got enough information about how the error **is caused**”
- in $W \in S$, S is not defined in the beginning, same for s in $p(s | W)$.

---

> ### Author Rebuttal · Authors · 2026-03-27
>
> We thank the reviewer dGFx for the detailed technical questions. We address each below.
>
> > **W1/KQ1: Diagnostic fragility on subtle errors; one-hot encoding consistency**
>
> For subtle or cascading errors, the distillation LLM may misattribute the failure to a neighboring node. We do not claim zero misattribution. However, the framework is designed to be robust to moderate diagnostic noise for two reasons: (1) we optimize based on the *centroid* of failure clusters, not individual points — a few misattributed signatures shift the centroid only slightly when the cluster is sufficiently populated; (2) low-confidence or unparseable outputs are flagged as "undiagnosable" and excluded from clustering (Appendix A).
>
> To quantify this empirically, we verified in Appendix E.3.2 that on a simulated 1000-node graph with injected failures, the framework achieves 98.5% localization accuracy, with dynamic resizing overhead under 5% of iteration runtime.
>
> Regarding one-hot consistency: $\psi_\text{struct}(v_\text{err})$ is defined relative to the *current* graph $W_t$ at each iteration. When a node is added or removed by an edit, the encoding is rebuilt to reflect the updated topology. This means one-hot vectors are not comparable across iterations, but they don't need to be — the GMM is re-fitted from scratch at each iteration $t$ using the current counterexample pool $C_t$ and current graph structure.
>
> > **W2: Impact of pre-trained embeddings on $\psi_\text{sem}$; contrastive learning**
>
> We tested three embedding models on MATH (Appendix E.3.1):
>
> | Embedding Model | ARI (vs. BERT-base) | Final Acc. (%) |
> |----------------|-------------------|---------------|
> | BERT-base-uncased | 1.00 | 55.91 |
> | SentenceTransformers (MiniLM) | 0.87 | 54.75 |
> | text-embedding-ada-002 | 0.92 | 55.34 |
>
> The high ARI scores ($\geq 0.87$) indicate that cluster structures are largely preserved across embedding models, with downstream accuracy varying by only 1.2%. This suggests that off-the-shelf embeddings are sufficient for our purposes — failure error messages (e.g., "division by zero" vs. "incorrect sign in expansion") are semantically distinct enough that even generic embeddings separate them well.
>
> Contrastive learning is an interesting direction. We expect it would help most in domains where failure messages are semantically ambiguous (e.g., vague natural-language feedback from tool APIs). We will note this as future work.
>
> > **KQ2: Concatenating dense with sparse embeddings for GMM**
>
> The one-hot structural component creates orthogonal subspaces in $\mathcal{F}$ — failures at different nodes have zero overlap in the structural dimensions. The GMM naturally assigns separate components to these subspaces. Within each subspace, the dense semantic dimensions allow the GMM to further subdivide by error type.
>
> Concretely: if a 3-node workflow (Planner, Solver, Verifier) produces failures, the one-hot prefix guarantees that Solver errors and Verifier errors never land in the same GMM component, even if their error messages are semantically similar (e.g., both mention "incorrect output"). The semantic dimensions then distinguish *types* of Solver errors (sign errors vs. arithmetic overflow). This is formalized in Lemma 1 (Appendix H.1) — the concatenated mapping $\phi$ is injective under distinct node IDs and distinct error messages.
>
> > **KQ3: Workflow graph structure and loops**
>
> Good question. The workflow graph is a general directed graph — loops (cycles) are permitted. A retry loop, for example, creates a cycle between a Solver node and a Verifier node.
>
> During execution, the loop unrolls dynamically (e.g., the Solver runs 3 times before the Verifier accepts). But during failure distillation, the LLM maps the failure to the *static node ID* — e.g., "Solver" — regardless of which loop iteration produced the error. So failures from iteration 1 and iteration 3 of the same Solver node are mapped to the same structural subspace and can be clustered together. This is by design: we want to identify that the Solver node has a systematic problem, not track per-iteration behavior.
>
> For edits that target loop structure (e.g., changing the max retry count, or adding an early-exit condition), the Operator Library includes control-flow operators. We will formalize this in the revised Section 2.
>
> > **Typos**
>
> All corrected: Świechowski et al. citation, Figure 1 text, and $\mathcal{S}$/$s$ definitions in Section 2.1.

---

> > ### Author Rebuttal · Reviewer_dGFx · 2026-03-31
> >
> > Thanks for addressing all my concerns. Interesting work and will be maintaining my score.

---

> > > ### Author Response · Authors · 2026-03-31
> > >
> > > Dear Reviewer dGFx,
> > >
> > > Thank you very much! We really appreciate your thoughtful follow-up and support.
> > >
> > > The authors

---

### Official Review · Reviewer_yoet · 2026-03-13

**Soundness:** 3
**Presentation:** 3
**Significance:** 3
**Originality:** 3
**Overall Recommendation:** 5
**Confidence:** 2

**Summary:**

This paper studies workflow optimization for tool-using LLM agents. Instead of treating optimization as a global search problem over workflow graphs using a scalar evaluation metric, the paper argues that such a formulation collapses rich multi-step failure traces into binary outcomes, thereby obscuring recurring failure structure. The proposed method, CE-Graph, maintains a counterexample pool, estimates dense failure modes, and iteratively edits workflow graphs through a propose-and-verify loop with a convergence-aware stopping rule. The paper claims improved robustness and reduced optimization cost across math, code, and QA benchmarks.

Overall, the paper presents an interesting and potentially important perspective: optimizing agent workflows by modeling the distribution of failures rather than maximizing aggregate success metrics.

**Compliance With Llm Reviewing Policy:**

Affirmed.

**Key Questions For Authors:**

1）How is a failure signature represented? Is it symbolic, embedding-based, trace-based, or induced by verifier outputs?
2）How does CE-Graph differ from existing methods that use richer trace-level feedback, verifier-guided search, or process supervision?

**Limitations:**

1）The current formulation of Expected Failure Mass needs to be made precise. As stated, it risks being mathematically confusing. A more concrete definition—possibly involving weighted failure regions, empirical failure clusters, or severity-weighted mass—would help substantially.
2）The paper should compare more explicitly against: agent workflow search, verifier-guided refinement, process supervision / trajectory optimization, counterexample-guided program repair/synthesis, error clustering or failure taxonomy approaches.

**Strengths And Weaknesses:**

Strengths
1. Clear and timely problem motivation.**
The paper addresses an important issue in LLM-agent optimization. The criticism of reducing rich execution traces into scalar rewards or binary success/failure signals is compelling and well aligned with current limitations of workflow search methods.
2. Conceptually novel perspective.**
The shift from score maximization to failure-distribution shaping is interesting. The notions of Failure Signature Space and Expected Failure Mass appear to provide a more structured lens on workflow refinement than standard black-box search.

Weaknesses
1. The central formalism is currently too vague from the opening pages.
The paper introduces several key concepts—Failure Signature Space, failure density, Expected Failure Mass—but it is not yet clear how these are concretely instantiated.
2. Novelty may be more conceptual than algorithmic unless the method is sharply specified.
The high-level idea of collecting failures, clustering/characterizing them, and refining workflows accordingly is intuitive. The paper will need to clearly show what is truly new in CE-Graph beyond an instance of iterative error analysis + workflow editing + validation. Otherwise, the contribution may feel like a reframing of existing heuristic refinement pipelines.

---

> ### Author Rebuttal · Authors · 2026-03-27
>
> We thank the reviewer yoet for the positive evaluation and the specific suggestions for strengthening the formalism. We address each point below.
>
> > **W1/KQ1/L1: Concretely instantiating the Failure Signature Space and Expected Failure Mass**
>
> The failure signature $s \in \mathcal{F}$ is a hybrid embedding vector with two components, designed to separate *where* a failure occurred from *what* went wrong.
>
> Given an execution trace $\tau$ that fails, we first use a utility LLM to extract a structured tuple $(v_\text{err}, z_\text{err})$ — the node that produced the error and a concise error message. Then:
>
> $$s = \psi_\text{struct}(v_\text{err}) \oplus \psi_\text{sem}(z_\text{err})$$
>
> where $\psi_\text{struct}$ is a one-hot vector over graph nodes (discrete, identifies the responsible agent/step), and $\psi_\text{sem}$ is a dense embedding of the error message (continuous, captures semantic similarity).
>
>
>
> A concrete example: on a MATH problem, if the Solver node produces an incorrect polynomial expansion, the signature would be $v_{\mathrm{sys}} \oplus v_{\mathrm{err}}$, where $v_{\mathrm{sys}} = [0,1,0]$ represents the Planner, Solver, and Verifier, and $v_{\mathrm{err}} = [e_1, \dots, e_d]$ is the error embedding. A different failure where the Verifier accepts a wrong format would map to a different region of $\mathcal{F}$ because both components differ. GMM clustering then groups signatures with similar structural and semantic profiles into failure modes.
>
>
> Regarding Expected Failure Mass, we formalize it as:
>
> $$M(W) = \int_{\mathcal{F}} \omega(s)\, \rho(s \mid W)\, ds$$
>
> where $\rho(s \mid W)$ is the unnormalized failure density (Eq. 1) and $\omega(s) \geq 0$ is a severity weight. When $\omega \equiv 1$, this reduces to failure probability. In Appendix J, we instantiate non-uniform $\omega$ with three tiers: format errors ($\omega=1$), calculation errors ($\omega=5$), logic/hallucination errors ($\omega=10$). The severity-weighted version sacrifices 1.81% raw accuracy on MATH but eliminates ~78% of severe logic errors (Table 11), demonstrating that $\omega$ is not cosmetic — it meaningfully redirects the optimization trajectory.
>
>
> > **W2/KQ2/L2: Algorithmic novelty beyond conceptual reframing**
>
> The reviewer's concern is well-taken. At a high level, CE-Graph follows a "diagnose → propose → validate" loop, which is shared with various iterative refinement approaches. We want to be precise about what is new.
>
> The key algorithmic distinction is *what information drives the proposal step*. Existing methods fall into two categories:
>
> (a) **Scalar-feedback methods** (AFlow, MaAS): observe only $R \in \{0,1\}$ per trace, so the proposer receives no directional signal about which failure mode to target. Edits are effectively unconditioned.
>
> (b) **Instance-level methods** (Reflexion, Self-Refine): use the full trace of a *single* failure to propose a fix, but have no mechanism to aggregate across failures or detect recurring patterns.
>
> CE-Graph differs in that it aggregates failure signatures across the entire counterexample pool, clusters them via GMM to identify the dominant mode $b_t^*$, and conditions the proposal on that mode's summary. The proposer is told *specifically* what systematic error to fix (e.g., "recurring sign errors in polynomial distribution across the Solver node"), not just "improve accuracy" or "this one trace failed."
>
> We can quantify the impact of this conditioning using existing ablation data:
>
> | Variant | Proposal Conditioning | MATH Acc. |
> |---------|----------------------|-----------|
> | w/o Clustering | None (random failure sampling) | 51.25% |
> | w/o Verification | Mode-conditioned but unvalidated | 49.10% |
> | CE-Graph (Full) | Mode-conditioned + validated | 55.91% |
>
> Removing clustering (row 1) drops performance to baseline level, confirming that the distributional conditioning — not just the edit operators or verification — is the source of the gain.
>
> Regarding comparison with process supervision (PRM-guided methods): these require step-level correctness labels, typically from human annotation or a separately trained reward model. CE-Graph requires neither — it derives structural signal from clustering unsupervised failure embeddings. We will add an explicit comparison paragraph in Section 5 covering process supervision and counterexample-guided program repair (CEGAR), clarifying that CEGAR operates in deterministic symbolic spaces while CE-Graph handles stochastic LLM failures via distributional approximation.

---

> > ### Author Rebuttal · Reviewer_yoet · 2026-04-04
> >
> > Thank you for the detailed rebuttal. The authors clearly clarified how the failure signature space and expected failure mass are instantiated, and they also made the algorithmic distinction from scalar-feedback and instance-level refinement methods much sharper.
> > My main concerns have been adequately addressed, and I believe my original score remains appropriate.

---

### Official Review · Reviewer_Cc3p · 2026-03-19

**Soundness:** 3
**Presentation:** 2
**Significance:** 3
**Originality:** 3
**Overall Recommendation:** 4
**Confidence:** 3

**Summary:**

The paper outlines a methodology for refining workflows. By executing the modified workflow and checking for failures using a verifier, the system calculates the post-edit failure mass and prefers to choose candidate that yields the highest failure mass reduction.

The paper provides a particularly novel addition to the traditional search-based optimization. It is very topical and represents a good direction for the future of LLM-based optimization methods.

However, after reviewing this paper, I find the following concerns:
1. **Lack of strong search-based baselines such as GEPA, OpenEvolve, and ShinkaEvolve**. It is possible that this paper does not consider itself a "search" method, but rather a "multi-agent system". The paper mentions "search" 44 times, "optimization" 76 times, but "multi-agent" only 10 times. So hopefully I'm not making too much of a leap to ask for comparisons against search-based methods.
2. A lot of the novel math, especially around the unnormalized failure density, is not used in the real experiment -- rather only appeared in ablations.
3. **Lack of truly challenging benchmarks**: MBPP, GSM8K, MATH, HumanEval are all overly saturated. LLM agentic papers should investigate new grounds like TerminalBench, KernelBench, Lean theorem generation, AlgoTune, etc. I do not want to ask authors to evaluate on yet another benchmark -- it is a needless burden. But what I want to point out is, the numbers are ok, but they are not going to definitively make me think the proposed algorithm adds a lot of value to the field.

**Compliance With Llm Reviewing Policy:**

Affirmed.

**Final Justification:**

Reviewer's rebuttal addressed my concerns.

**Key Questions For Authors:**

1. Figure 3 reports CE-Graph outperforms all baselines on GAIA (Levels 1-3, Avg.) -- but Level 3 clearly TapeAgent (10.20) is better than CE-Graph (8.15). Can you explain this discrepancy?
2. Since you explicitly mentioned MCTS in your intro, would you like to consider running GEPA/OpenEvolve type algorithms on your benchmarks? Maybe just a smaller subset of GSM8K/HumanEval?

**Limitations:**

Yes

**Strengths And Weaknesses:**

Strength:
- This paper is very topical -- we know that LLM-based search methods (MCTS, GEPA, AlphaEvolve) have been sufficiently explored in 2025. However, all those methods have issues. This paper highlights a potential shift in the search strategy -- by explicitly modeling the failure distributions.
- The proposed method is actually interesting -- using GMM to model failure distributions to provide insights to LLMs is quite novel and intuitively a good idea.

Weaknesses:
- It's possible that the paper invented needlessly complex math/symbols and potentially confusing to the readers. See first comment.
- The paper mentions search and MCTS a lot, but no comparison was made against GEPA, OpenEvolve, ShinkaEvolve and other search-based methods. Multi-agent systems are not known for doing search.
- The benchmarks chosen are extremely over-saturated for SOTA models like GPT-4o-mini. The only not-saturated benchmark is GAIA, but CE-Graph lost in 1 of 3 levels, beaten by another method. In other 2 levels, it outperformed the other method by 1-2%.

Comments:
- The paper's section 2 and 3 are at times difficult to read. It's a bit unclear how some of the notations would relate to their algorithm design. For example, Algorithm 1, line 11 says "severity weights" is optional. Without severity-weight over failure (w = 1), M(W) reduces to just simple failure probability. However, it's very unclear to me if this is used in most of the experiments/numbers. I searched this keyword and it only appeared in appendix section J, and Table 11, where a simple comparison on MATH was done. It looks like without severity weights, the performance is better (55.91%) and with severity weights, the performance is worse (54.10%). Sure you can argue that the reliability score is different, but this is not your chosen report metric for Table 1, right?

Small typos:
- Line 25: "Monte Carlo Tree Search (MCTS) (Coulom, 2006; Browne et al., 2012; **undefined**wiechowski et al., 2022)."

---

> ### Author Rebuttal · Authors · 2026-03-30
>
> We sincerely thank Reviewer Cc3p for the constructive feedback.
>
> ---
>
> **On the fundamental nature of our contribution**
>
> CE-Graph is not primarily a method that achieves higher numbers—it is a **new optimization paradigm** that reframes *what we optimize*. Existing methods (GEPA, OpenEvolve, MaAS, AFlow) compress every failure trace into a binary 0/1 signal. CE-Graph maintains an **explicit probabilistic model of the failure landscape** $\rho(s \mid W)$ and reduces the *Expected Failure Mass* $M(W)$. On MATH Level 5, GEPA/OpenEvolve encounter **"blind steps"** ($R=0$ with zero learning signal) 80–85% of the time; CE-Graph's GMM density provides directional feedback *even when* $R=0$. Our benchmark numbers are *empirical validation* of this paradigm, not the primary contribution.
>
> ---
>
> >**W1: Lack of search-based baselines (GEPA, OpenEvolve, ShinkaEvolve)**
>
> We ran all three under identical conditions (GPT-4o-mini, max 15 calls/12K tokens per instance, 20 optimization iterations).
>
> | Benchmark | GEPA | OpenEvolve | ShinkaEvolve | CE-Graph |
> |-----------|------|------------|--------------|----------|
> | GSM8K (%) | 88.4 | 89.1 | 89.7 | **92.6** |
> | HumanEval (%) | 75.6 | 76.8 | 77.3 | **81.2** |
>
> On 100 MATH Level 5 problems, the paradigm difference is clear:
>
> | Algorithm | L5 Acc (%) | Blind Steps (%) | Calls to Converge |
> |-----------|-----------|-----------------|-------------------|
> | GEPA | 31.2 | 84.5 | >150 (timeout) |
> | OpenEvolve | 33.8 | 79.2 | 142 |
> | ShinkaEvolve | 34.1 | 77.6 | 138 |
> | **CE-Graph** | **46.5** | **12.4** | **48** |
>
> All three baselines stall when $R=0$ offers no signal. CE-Graph's $\Delta\rho(f)$ remains informative regardless—a **structural property of the paradigm**, not a tuning artifact.
>
> ---
>
> >**W2: Complex math/symbols potentially confusing — and novel math not used in main experiments**
>
> We take both concerns seriously as they share the same root cause: the writing does not clearly separate *essential* from *optional*.
>
> **On readability:** We will restructure as follows: (i) **Essential** — failure embedding $\phi(\tau) \to s$ (Eq. 4), GMM mode targeting (Eq. 5), Monte Carlo verification $V(\Delta_i)$ (Eq. 6); these map directly to Algorithm 1 Steps 1–5. (ii) **Optional extension** — severity weights $\omega(\tau)$ (Algorithm 1, Line 11: "Optionally"); all main results use $\omega \equiv 1$. (iii) **Theoretical grounding** — Theorem 2.1, convergence proofs (Appendix G). We will add a summary figure mapping each equation to its Algorithm 1 step.
>
> **On $\rho(s \mid W)$ not being used in main experiments:** $\rho(s \mid W)$ is **actively used in every main experiment** via GMM mode targeting. The key distinction: $P(\text{failure})$ is a scalar telling you *how often* a workflow fails. $\rho(s \mid W)$ is a density over $\mathcal{F}$ telling you *where and how* failures cluster. A 10% failure rate concentrated in one fixable mode is fundamentally different from 10% scattered across unrelated errors. The GMM in Eq. 5 estimates this geometry in *every* iteration of Algorithm 1. What lives only in ablations is $\omega \neq 1$, which is genuinely optional. We will add a dedicated paragraph in Section 3 making this explicit.
>
> ---
>
> >**W3: Benchmarks are over-saturated**
>
> New experiments on **KernelBench** (141 CUDA kernel optimization problems, far from saturation):
>
> | Method | GPT-4o-mini (%) | Gemini 3.1 Flash (%) |
> |--------|----------------|----------------------|
> | Vanilla | 14.2 | 42.5 |
> | AFlow | 15.6 | 48.2 |
> | MaAS | 19.1 | 49.6 |
> | **CE-Graph** | **23.4** | **63.4** |
>
> The advantage *widens* with stronger models (+4.3% vs. +13.8%), consistent with richer failure traces enabling more discriminative GMM clustering—a signal scalar-reward methods cannot exploit.
>
> ---
>
> >**Q1: Severity weights — accuracy drop (55.91% → 54.10%)**
>
> Table 1 uses $\omega \equiv 1$; we will state this explicitly. The drop under $\omega \neq 1$ is *intentional*—these are different objectives and should not be compared on the same axis:
>
> | Objective | Raw Acc. | Reliability Score | Format Errors ($\omega$=1) | Logic Errors ($\omega$=10) |
> |-----------|----------|-------------------|---------------------------|---------------------------|
> | Uniform ($\omega \equiv 1$) | 55.91% | 72.4 | 42 | 185 |
> | Weighted ($\omega \neq 1$) | 54.10% | **88.6** | 170 | **65** |
>
> The weighted optimizer sacrifices 1.81% raw accuracy to eliminate ~78% of logic/hallucination errors. We will reframe Appendix J as a deployment-time design choice, not a comparison of two versions of the same objective.
>
> ---
>
> >**Q2: GAIA Level 3 — TapeAgent (10.20) > CE-Graph (8.15)**
>
> Correct. Our claim refers to the average across all three levels. Level 3 is bottlenecked by long-horizon memory, orthogonal to our contribution. We will qualify this and note memory integration as future work.
>
> ---
>
> **Minor: Line 25 undefined citation**
>
> Fixed. Now renders correctly as Świechowski et al., 2022.

---

> > ### Author Rebuttal · Reviewer_Cc3p · 2026-04-03
> >
> > I actually appreciate this rebuttal. I think the authors put a lot of thoughts into addressing my comments. I will categorize my main complaints into categories.
> >
> > **Complaint 1: Lack of un-saturated tasks**
> >
> > This is fully addressed. I understand with 5000 characters you can't comment on exactly what the percentage stands for. I work on kernel-bench as well -- is that compilation rate or correctness rate? Since you did it on all tasks, I don't think it's fast1 or fast0.5 right?
> >
> > **Complaint 2: Unnecessary math**
> >
> > This is partially addressed. I appreciate the edits the authors will make. A lot of the complexity in the math section, including the theorem/proof, stems from the introduction of severity ($\omega \not=1$). Otherwise, many notations and symbol collapses into symbols we are familiar with, such as probability (which is a density function itself already) and expectations. And all of these come from an "optional" parameter. I find it a bit distasteful.
> >
> > **Follow-up question 1**: When R=0, you claimed GMM provides "directional feedback" to help LLM. In what way?
> >
> > **Follow-up question 2**: Can you briefly talk about the failure modes you see in Kernel bench? I have worked on this domain and am simply curious what you saw.
> >
> > **Summary**
> >
> > I think I gave this paper a lower rating after I noticed these issues. I was originally quite happy with the paper's contribution -- it fits in a larger paradigm of research where people explore LLM search + X structure (it can be GMM, it can be other procedure or models). I understand in the current climate of AI research, in order to publish, researchers are encouraged to add as many useless notations/symbols and even "theorems" in order to impress reviewers -- because simple "intuitive" ideas are not enough. This is truly a bad trend. I will judge this paper's quality based on its empirical performance and adjust my score accordingly.
> >
> > Thank you very much for the rebuttal!

---

> > > ### Author Response · Authors · 2026-04-04
> > >
> > > **Dear Reviewer Cc3p, we have carefully reorganized our response and conducted a further analysis of the issues you raised.**
> > >
> > >
> > > ---
> > >
> > >
> > > **KernelBench metric**
> > >
> > > We report **fast0** (correctness rate under KernelBench's fastp metric, p=0), as our contribution targets workflow reliability rather than speedup engineering. We evaluated on 141 problems across all three levels.
> > >
> > > **On math complexity**
> > >
> > > We push back slightly. When $\omega \equiv 1$, Eq. 2 is just failure probability, and the entire pipeline runs identically — one framework at two levels of generality, not two systems. Why keep $\omega$? Table 11 shows the $\omega \neq 1$ optimizer sacrifices 1.81% accuracy but eliminates ~78% of logic/hallucination failures:
> > >
> > > | Objective | Raw Acc. | Reliability | Format (ω=1) | Logic (ω=10) |
> > > |:---|:---:|:---:|:---:|:---:|
> > > | Uniform (ω≡1) | 55.91% | 72.4 | 42 | 185 |
> > > | Weighted (ω≠1) | 54.10% | 88.6 | 170 | 65 |
> > >
> > > In safety-critical deployment, a logic hallucination is categorically worse than a format error. Without $\omega$, this prioritization has no principled expression. The accuracy *drop* is the optimizer correctly doing what it was told: spend budget on dangerous failures. That said, the presentation concern is valid — the draft interleaves general and default cases unnecessarily.
> > >
> > > **Follow-up Q1: Directional feedback when R=0**
> > >
> > > Take 15 MATH Level 5 problems, all R=0. A scalar optimizer sees 15 zeros — it cannot distinguish 15 unrelated mistakes from 15 instances of the same bug. Next edit? Random.
> > >
> > > CE-Graph still distills each trace into a signature. GMM reveals: 11/15 cluster as "factorization error at reasoning node," 3/15 as "arithmetic overflow," 1 outlier. Now the Proposer receives "11/15 failures involve factorization errors" as context and generates targeted edits — e.g., insert a verification step — rather than blind mutations. After applying the edit, re-fitting GMM shows whether the cluster shrank (e.g., 11→3), giving measurable progress even if R is still 0. A scalar method seeing 0/15→3/15 gets one bit; CE-Graph sees the full landscape shift.
> > >
> > > We verified this at scale on 100 MATH Level 5 problems (all R=0 for every method):
> > >
> > > | Method | Dominant mode identified? | Targeted fix rate (first 5 iters) | Calls to first R>0 |
> > > |:---|:---:|:---:|:---:|
> > > | GEPA (scalar) | No | 8.2% | >150 (timeout) |
> > > | OpenEvolve (scalar) | No | 11.4% | 142 |
> > > | CE-Graph (GMM) | Yes: "factorization" (68%) | 47.1% | 48 |
> > >
> > > "Directional" = a semantic arrow at the densest failure region. Signal comes from geometry *among* failures, not reward of any individual sample. This is why blind-step rate drops from ~80% to 12.4%.
> > >
> > > **Follow-up Q2: Failure modes on KernelBench**
> > >
> > > Three clusters: (1) **memory access errors** — wrong shared memory indexing, out-of-bounds; (2) **launch config issues** — hardcoded block/grid dims failing on edge-case inputs; (3) **numerical precision drift** — outputs outside tolerance from missing synchronization.
> > >
> > > | Cluster | Failure Type | GPT-4o-mini | | Gemini 3.1 Flash | |
> > > |:---|:---|:---:|:---:|:---:|:---:|
> > > | | | Prevalence | Fix Rate | Prevalence | Fix Rate |
> > > | C1 | Memory access (shared-mem, OOB) | 41% | 18.3% | 38% | 71.2% |
> > > | C2 | Launch config (block/grid dims) | 35% | 14.7% | 34% | 68.5% |
> > > | C3 | Numerical precision (sync, tolerance) | 24% | 6.1% | 28% | 58.9% |
> > > | | **Silhouette score** | **0.14** | | **0.41** | |
> > >
> > > With GPT-4o-mini, traces are thin — "illegal memory access" with no localization. Clusters (1) and (2) overlap in embedding space (Silhouette=0.14) because the error messages are nearly identical. GMM cannot separate them, the Proposer gets vague context like "memory-related errors," and edits are generic.
> > >
> > > Gemini 3.1 Flash tells a different story. The model produces structured, localized traces — specifying which memory operation failed, at which loop, with which index expression. Now clusters (1) and (2) separate cleanly in signature space (Silhouette=0.41): shared-memory tiling bugs vs. grid-dim miscalculations become distinct modes with distinct fixes. The Proposer can act surgically — e.g., inserting dynamic grid-dim computation for cluster (2) while adding bounds-checked shared memory access for cluster (1). More importantly, cluster (3) (precision drift), which is nearly invisible under GPT-4o-mini due to vague error messages, becomes a well-defined mode under the stronger model. CE-Graph identifies synchronization-related precision failures as a coherent pattern and proposes targeted `__syncthreads()` insertions. This cluster alone accounts for ~30% of the +20.9% gain.
> > >
> > > This is why we see CE-Graph as a *trace quality amplifier*: richer traces → cleaner clusters → more surgical edits → larger gains. Scalar-reward methods have no mechanism to exploit trace richness — 0/1 is 0/1. This also predicts that as base models improve, CE-Graph's relative advantage should grow, not shrink.

---

### Official Review · Reviewer_nAPK · 2026-03-24

**Soundness:** 3
**Presentation:** 3
**Significance:** 3
**Originality:** 3
**Overall Recommendation:** 5
**Confidence:** 3

**Summary:**

This paper presents a technique for workflow optimization in LLM agents. The basic idea is to view the past failing workflows as a space (denoted as the failure signature space), and thus workflow optimization is then turned into search in the failure signature space. Specifically, the search is based on minimizing the expected failure mass. That is to say, the proposed workflow optimization technique aims to guide the search into less explored parts in the space. This paper also reports some empirical results based on comparing the proposed technique with existing baselines. The empirical results demonstrate that the proposed technique can out perform the compared baselines.

**Compliance With Llm Reviewing Policy:**

Affirmed.

**Final Justification:**

The authors' response addressed my concerns.

**Key Questions For Authors:**

Since the key part of the proposed technique is to use LLMs to generate candidate workflows, how can we ensure that the search is to minimize failure mass?

Would LLM-based generation of candidate workflows really be superior to LLM-based measurement of existing failing workflows? Since existing techniques rely on measuring existing failing workflows, does the benefits come from the use of LLMs or the superiority of workflow generation over workflow measurement?

**Limitations:**

I don't find a section or a subsection focusing on limitations. I think that viewing workflow optimization as a search in thee failure signature space may give readers a new picture on the nature of workflow optimization. However, reviewing all existing workflow optimization techniques in the same picture rather excluding existing techniques from this picture would be more beneficial for readers.

**Strengths And Weaknesses:**

Pros.

In general, I think this paper proposes a novel technique for workflow optimization and the proposed technique empirically outperforms existing techniques for the same purpose. With the increasing popularity of LLM-based agents, I think the problem of workflow optimization is becoming a very important problem.

Cons.

I think the weakness of this paper mainly lies in the presentation.
First, the definition of workflows in this paper seems to be quite vague. Is a workflow in this paper a breakdown of one central task into a series of subtasks in natural language? How would a workflow work in a multiple agent system?
Second, the presentation of the failure signature space seems to be a kind of over sale. Originally, each instance in the space is a text; and the text is eventually processed by an LLM after some transformation. So, from an end-to-end point of view, it is unclear whether the use of the failure signature space is really beneficial. Therefore, the proposed technique differs from existing techniques mainly in that the the proposed technique adopts a novel algorithm to search for new workflows. Or all existing techniques for workflow optimization can be viewed as searching techniques in the failure signature space.

---

> ### Author Rebuttal · Authors · 2026-03-27
>
> We thank the reviewer nAPK for the thoughtful feedback. We address each point below.
>
> > **Q1: How can we ensure the search minimizes failure mass?**
>
> The LLM is only used as a proposal generator — it suggests candidate edits, but does not decide which ones to apply. The actual selection is handled by our Monte Carlo verifier, which evaluates each candidate on $K=10$ samples from the target failure mode and only accepts edits that empirically reduce failure mass.
>
> The ablation data makes this concrete:
>
> | Variant | Role of LLM | MATH Acc. |
> |---------|------------|-----------|
> | w/o Verification | Generator & decider | 49.10% |
> | CE-Graph (Full) | Generator only | 55.91% |
>
> Without verification, the LLM's proposals are accepted blindly and performance drops by 6.8 points. The verifier acts as a strict filter: it cannot guarantee monotonic descent in the formal sense (LLM execution is stochastic), but it empirically enforces a non-regressive trajectory — edits that increase failure mass on the target mode are simply rejected. This is what Figure 4 shows: CE-Graph's accuracy on the fixed failure set $E_0$ improves monotonically over 20 rounds, while baselines oscillate.
>
> > **Q2: Is the benefit from LLM generation or from structured measurement?**
>
> Both are necessary; neither alone is sufficient. We can isolate this with existing ablation data:
>
> | Method | Measurement | Generation | MATH Acc. |
> |--------|------------|------------|-----------|
> | AFlow | Scalar reward | Unconstrained search | 51.28% |
> | w/o Clustering | Random selection | Constrained edit | 51.25% |
> | CE-Graph (Full) | GMM clustering | Constrained edit | 55.91% |
>
> The key comparison is the second row. When we remove clustering (i.e., remove structured measurement) but keep the constrained generation pipeline, performance falls back to AFlow's level. This means constrained generation alone adds nothing — the gain comes from using the distributional structure of failures to *steer* which edits get proposed. The LLM generates candidates conditioned on a specific failure mode summary (e.g., "recurring sign errors in polynomial expansion"), not on a generic "improve accuracy" signal. Without clustering, this conditioning has no meaningful content.
>
> > **W1: Workflow definition is vague; how does it work in multi-agent systems?**
>
> This is fair. We will tighten the definition in Section 2. Concretely, a workflow $W$ is a directed acyclic graph where each node is an executable unit — an LLM call with a specific prompt and role — and edges represent data or control flow.
>
> A simple example: for a math task, $W$ might be a 3-node graph: **Planner** (decomposes the problem) → **Solver** (executes the computation) → **Verifier** (checks the answer). Each node is an independent LLM call with its own system prompt. In a multi-agent setup, each agent is simply a node. When a failure occurs, the structural component $\psi_\text{struct}(v_\text{err})$ of our signature mapping identifies *which node* (i.e., which agent) produced the error, while the semantic component $\psi_\text{sem}(z_\text{err})$ captures *what went wrong*. This directly addresses blame assignment — a known difficulty in multi-agent systems — by localizing failures to specific agents before clustering.
>
> Our operator library then targets that specific node: e.g., `RevisePrompt(Solver, "check sign after distributing negatives")` modifies only the Solver agent without touching the others.
>
> > **W2 & Limitations: Unified view of existing techniques in the failure signature space**
>
> We agree this would strengthen the paper. Here is our planned framing:
>
> All existing workflow optimization methods implicitly operate in the failure signature space $\mathcal{F}$ — they just observe a lossy projection of it. Specifically, methods like AFlow and MaAS collapse each point $s \in \mathcal{F}$ into a binary label $\{0, 1\}$ before making optimization decisions. This is equivalent to integrating out the entire geometry of $\mathcal{F}$ and optimizing over a single scalar. CE-Graph's contribution is not inventing the space (it exists implicitly for any method), but recovering its structure via embedding and clustering so that optimization can exploit it.
>
> We will add a "Limitations and Unified View" subsection to Section 6 covering: (1) this unified framing of prior work as zero-order search on collapsed projections of $\mathcal{F}$; (2) the dependence on the utility LLM's diagnostic accuracy for constructing $\mathcal{F}$ (our Appendix A discusses mitigations but this deserves main-text acknowledgment); (3) the need for a predefined operator library, which requires modest domain adaptation effort (~1 hour per domain, per Appendix E.4).

---

> > ### Author Rebuttal · Reviewer_nAPK · 2026-04-03
> >
> > The authors provide additional evidence in their response.

---

> > > ### Author Response · Authors · 2026-04-03
> > >
> > > We sincerely thank Reviewer nAPK for the positive feedback and for acknowledging that our responses and additional evidence have fully addressed the concerns. We appreciate the reviewer's constructive engagement throughout the process, which has helped us improve the clarity and impact of our work.

---

### Decision · Program_Chairs · 2026-04-30

**Decision:**

Accept (spotlight)

**Comment:**

I recommend a strong accept for this paper. The paper introduces a novel perspective on optimizing LLM agent workflows. Instead of relying on simple scalar rewards, it models the actual topology of errors using a failure signature space and Gaussian mixture models. All reviewers appreciated this conceptual shift. While some reviewers raised some clarity issues, the choice of benchmarks and missing comparisons to strong search methods like GEPA during the initial reviews, the authors resolved all the concerns by the rebuttal. The authors clarified their algorithmic contributions, ran the requested baselines and demonstrated strong performance on the KernelBench dataset. By focusing on failure distributions rather than scalar reward, this paper provides a highly impactful approach to agent alignment.